# Apolipoprotein A-IV binds αIIbβ3 integrin and inhibits thrombosis

Xiaohong Ruby Xu[1,2,3,4], Yiming Wang[1,2,5], Reheman Adili[2], Lining Ju [6,7,8], Christopher M. Spring[2], Joseph Wuxun Jin[2,5], Hong Yang[2,5], Miguel A.D. Neves[2], Pingguo Chen[2,5], Yan Yang[2,5], Xi Lei[2], Yunfeng Chen[7,9], Reid C. Gallant[1,2], Miao Xu[1,2], Hailong Zhang[2], Jina Song[2,5], Peifeng Ke[4,10], Dan Zhang[2,4], Naadiya Carrim[2,5], Si-Yang Yu[2,11], Guangheng Zhu[2], Yi-Min She[12], Terry Cyr[12], Wenbin Fu[3,4], Guoqing Liu[13], Philip W. Connelly[1,2], Margaret L. Rand[1,14], Khosrow Adeli[1,15], John Freedman[1,2,16], Jeffrey E. Lee[1], Patrick Tso[17], Patrizia Marchese[18], W. Sean Davidson[17], Shaun P. Jackson[8,18], Cheng Zhu [6,7,9], Zaverio M. Ruggeri [18] & Heyu Ni[1,2,5,16,19]

Platelet αIIbβ3 integrin and its ligands are essential for thrombosis and hemostasis, and play key roles in myocardial infarction and stroke. Here we show that apolipoprotein A-IV (apoA-IV) can be isolated from human blood plasma using platelet β3 integrin-coated beads. Binding of apoA-IV to platelets requires activation of αIIbβ3 integrin, and the direct apoA-IV-αIIbβ3 interaction can be detected using a single-molecule Biomembrane Force Probe. We identify that aspartic acids 5 and 13 at the N-terminus of apoA-IV are required for binding to αIIbβ3 integrin, which is additionally modulated by apoA-IV C-terminus via intra-molecular interactions. ApoA-IV inhibits platelet aggregation and postprandial platelet hyperactivity. Human apoA-IV plasma levels show a circadian rhythm that negatively correlates with platelet aggregation and cardiovascular events. Thus, we identify apoA-IV as a novel ligand of αIIbβ3 integrin and an endogenous inhibitor of thrombosis, establishing a link between lipoprotein metabolism and cardiovascular diseases.

---

[1] Department of Laboratory Medicine and Pathobiology, University of Toronto, Toronto, ON, Canada M5S 1A1. [2] Department of Laboratory Medicine, Keenan Research Centre for Biomedical Science, Li Ka Shing Knowledge Institute, St. Michael's Hospital, and Toronto Platelet Immunobiology Group, Toronto, ON, Canada M5B 1W8. [3] Department of Acupuncture and Moxibustion, Guangdong Provincial Hospital of Chinese Medicine, Guangzhou, Guangdong, P.R. China 510120. [4] Guangzhou University of Chinese Medicine, Guangzhou, Guangdong, P.R. China 510000. [5] Canadian Blood Services Centre for Innovation, Toronto, ON, Canada M5G 2M1. [6] Coulter Department of Biomedical Engineering, Georgia Institute of Technology, Atlanta, GA, USA 30332. [7] Petit Institute for Bioengineering and Biosciences, Georgia Institute of Technology, Atlanta, GA, USA 30332. [8] Heart Research Institute, and Charles Perkins Centre, The University of Sydney, Camperdown, Australia 2006. [9] Woodruff School of Mechanical Engineering, Georgia Institute of Technology, Atlanta, GA, USA 30332. [10] Department of Laboratory Medicine, Guangdong Provincial Hospital of Chinese Medicine, Guangzhou, Guangdong, P.R. China 510120. [11] Department of Cardiovascular Medicine, The Second Xiangya Hospital of Central South University, Changsha, Hunan, P.R. China 410011. [12] Centre for Biologics Research, Biologics and Genetic Therapies Directorate, HPFB, Health Canada, Ottawa, ON, Canada K1A 0M2. [13] Institute of Cardiovascular Science, Peking University Health Science Center, Beijing, P.R. China 100083. [14] Division of Haematology/Oncology, The Hospital for Sick Children, Toronto, ON, Canada M5G 1X8. [15] Program in Molecular Structure & Function, The Hospital for Sick Children, Toronto, ON, Canada M5G 1X8. [16] Department of Medicine, University of Toronto, Toronto, ON, Canada M5S 1A1. [17] Department of Pathology and Laboratory Medicine, University of Cincinnati, Cincinnati, OH, USA 45219. [18] Department of Molecular and Experimental Medicine, The Scripps Research Institute, La Jolla, CA, USA 92037. [19] Department of Physiology, University of Toronto, Toronto, ON, Canada M5S 1A1. These authors contributed equally: Xiaohong Ruby Xu, Yiming Wang, Reheman Adili, Lining Ju, Christopher M. Spring, Joseph Wuxun Jin. Correspondence and requests for materials should be addressed to H.N. (email: nih@smh.ca)

Apolipoprotein A-IV (apoA-IV) is a 46 kDa exchangeable plasma protein that shares structural features with other apolipoproteins[1]. It is synthesized in the small intestine and can rapidly increase 3–5 fold in response to the absorption of dietary or biliary fat, particularly unsaturated fats[2,3]. After secretion into the intestinal lymphatics, apoA-IV primarily associates with chylomicrons and enters the blood circulation via the thoracic duct. Unlike other apolipoproteins (e.g., apolipoprotein A-I), apoA-IV binds to lipoprotein particles weakly and can be readily displaced by other apolipoproteins[4]. As a result, approximately 80% of circulating apoA-IV is lipid-free[3,5]. Although abundant in the blood (150–370 µg/mL)[6], the exact function of apoA-IV is still unclear. Roles postulated for apoA-IV to date include regulation of lipoprotein metabolism and reverse cholesterol transport, anti-oxidation, anti-inflammation, and control of food intake[3,7]. Several studies in different human populations have consistently demonstrated that the plasma levels of apoA-IV are inversely correlated with cardiovascular diseases (CVDs)[8–10], but its roles in platelet activity and thrombosis, the major cause of heart attack and stroke, and the leading cause of mortality and morbidity worldwide[11,12], are unknown.

Platelets are anucleate cells in the blood that play a key role in thrombosis and hemostasis[13–16]. They are also actively involved in inflammation[17,18], immune responses[19–21], tumor metastasis[22,23], and contribute to the initiation of atherosclerosis through interaction with other cells including endothelial cells, leukocytes, and endothelial progenitor cells[24,25]. Recently, the role of platelets in atherosclerosis has been highlighted since increased platelet production accelerates atherogenesis[25]. Upon rupture of the atherosclerotic lesion, platelet adhesion, and subsequent aggregation at the site of injury may lead to thrombosis and vessel occlusion, resulting in myocardial and/or cerebral infarction.

It has been well documented that platelet integrin αIIbβ3 is the dominant integrin expressed on platelets, which plays a key role in platelet adhesion and is required for platelet aggregation[26,27]. Although fibrinogen (Fg), a major ligand of platelet αIIbβ3, was considered to be essential for platelet aggregation (and contributing in part to platelet adhesion), Fg-independent thrombosis occurs[27–31]. This suggests that other ligands of αIIbβ3 exist which are involved in platelet aggregation and thrombosis, but little information is available regarding what they are and how they regulate thrombosis and hemostasis; two opposing but critical biological processes.

Here we show that apoA-IV is a novel ligand of platelet αIIbβ3 integrin. Through competing with Fg and other prothrombotic ligands, it attenuates platelet aggregation, thrombosis, and postprandial platelet hyperactivity, and is an important endogenous protective factor against CVDs.

## Results

### ApoA-IV is a novel ligand of αIIbβ3 integrin.
To search for the unknown ligands of αIIbβ3 integrin, we coated latex beads with RGD (arginine-glycine-aspartic acid) peptide-activated human platelet β3 integrins, and incubated them with human blood plasma. Protein(s) associated with β3 integrin-coated beads were electrophoresed and apoA-IV was identified by mass spectrometry. We found that apoA-IV bound to the surface of platelets activated by adenosine diphosphate (ADP) or collagen, but not to quiescent or β3 integrin-deficient platelets in platelet-rich plasma (PRP) (Fig. 1a). Furthermore, apoA-IV-platelet association was completely blocked by a specific anti-β3 integrin monoclonal antibody (mAb) M1[30] (Fig. 1a). Thus, binding of apoA-IV to platelets is dependent on activated β3 integrin.

Interestingly, apoA-IV bound to activated platelets, but was not internalized by β3 integrins since no apoA-IV was detected in resting platelets (Fig. 1b), which is different from either Fg or fibronectin[29,32]. To confirm whether apoA-IV is a ligand of β3 integrin, we generated biotinylated recombinant apoA-IV and detected the binding between apoA-IV and platelet β3 integrins using a Biomembrane Force Probe (BFP)[33,34] (Fig. 1c–e, Supplementary Fig. 1a, Methods). The BFP brings an αIIbβ3 integrin-coated bead or a platelet target into contact with an apoA-IV bearing force probe (Supplementary Fig. 1a), and detects the single-molecular adhesions from the pico-force signal measured upon target retraction (Supplementary Fig. 1b), while a zero force indicates a non-adhesion event[33] (Supplementary Fig. 1c). At 1 mM Ca$^{2+}$, apoA-IV bound to ADP-treated platelets with high frequency and had minimal interactions with resting platelets (Fig. 1c). ApoA-IV also bound to Chinese hamster ovary (CHO) cells that express αIIbβ3, but not to the cell lines that express GPIb-IX or αMβ2 integrin (Fig. 1d). Furthermore, at 1 mM Ca$^{2+}$, apoA-IV bound to purified αIIbβ3, but not αVβ3 and α5β1 integrins (Fig. 1e). Importantly, apoA-IV and αIIbβ3 interactions in these experiments were completely blocked by anti-β3 integrin mAb M1 (Fig. 1c–e). Kinetic measurements further showed that the effective two-dimensional (2D) affinity of apoA-IV-αIIbβ3 is approximately 40% of the affinity between Fg and αIIbβ3 (Fig. 2a). Notably, recombinant apoA-IV competitively inhibited the Fg-αIIbβ3 interaction in a dose-dependent manner in BFP (Fig. 2b), enzyme-linked immunosorbent assay (Fig. 2c) and a flow cytometric assay (Fig. 2d). Fg also inhibited apoA-IV-αIIbβ3 interaction (Supplementary Fig. 1d–e), indicating that apoA-IV and Fg inhibit one another in a reciprocal fashion. Furthermore, recombinant apoA-IV decreased Fg-mediated platelet adhesion (Supplementary Fig. 2a-b). These findings suggest that the occupancy of αIIbβ3 by apoA-IV inhibits Fg-αIIbβ3 interactions. We further found that RGD peptides, the canonical integrin binding motif, decreased apoA-IV binding to αIIbβ3 in a dose-dependent manner (Supplementary Fig. 3). Thus, apoA-IV is identified as a novel ligand of platelet integrin αIIbβ3.

### ApoA-IV inhibits platelet aggregation in vitro.
Fg-αIIbβ3 interaction plays an important role in platelet aggregation (Supplementary Fig. 4) and is required for platelet aggregation in anticoagulated blood[27]. Since apoA-IV can inhibit Fg-αIIbβ3 interaction, we first examined the role of apoA-IV in platelet aggregation using apoA-IV-deficient (apoA-IV$^{-/-}$) mice[6] and wild-type (WT) littermate controls (apoA-IV$^{+/+}$). Platelet aggregation was significantly enhanced in the PRP of apoA-IV$^{-/-}$ mice following stimulation with various agonists including ADP, collagen, and thrombin receptor activating peptide (TRAP, AYPGKF-NH$_2$) (Fig. 3a). To distinguish whether this enhancement was due to plasma apoA-IV deficiency, but not alteration of platelet function, we switched apoA-IV$^{-/-}$ and apoA-IV$^{+/+}$ plasma (platelet poor plasma, PPP), and found the enhancement of platelet aggregation was specific to plasma apoA-IV deficiency (Fig. 3b). This was further confirmed using plasma-free gel-filtered platelets, in which plasma was removed during Sepharose 2B column filtration, and no differences were observed in platelet aggregation between apoA-IV$^{-/-}$ and apoA-IV$^{+/+}$ mice (Fig. 3c). This is consistent with our data demonstrating that apoA-IV was not internalized by platelets (Fig. 1b) and therefore does not affect platelet granule release-related platelet aggregation. Considering the anti-inflammatory roles of apoA-IV[7], possible alterations of inflammatory factors and other plasma proteins in apoA-IV$^{-/-}$ mice may alter platelet aggregation, thus recombinant human and mouse apoA-IV were used directly to confirm these results. We found that recombinant apoA-IV inhibited both WT mouse and human ADP-induced platelet

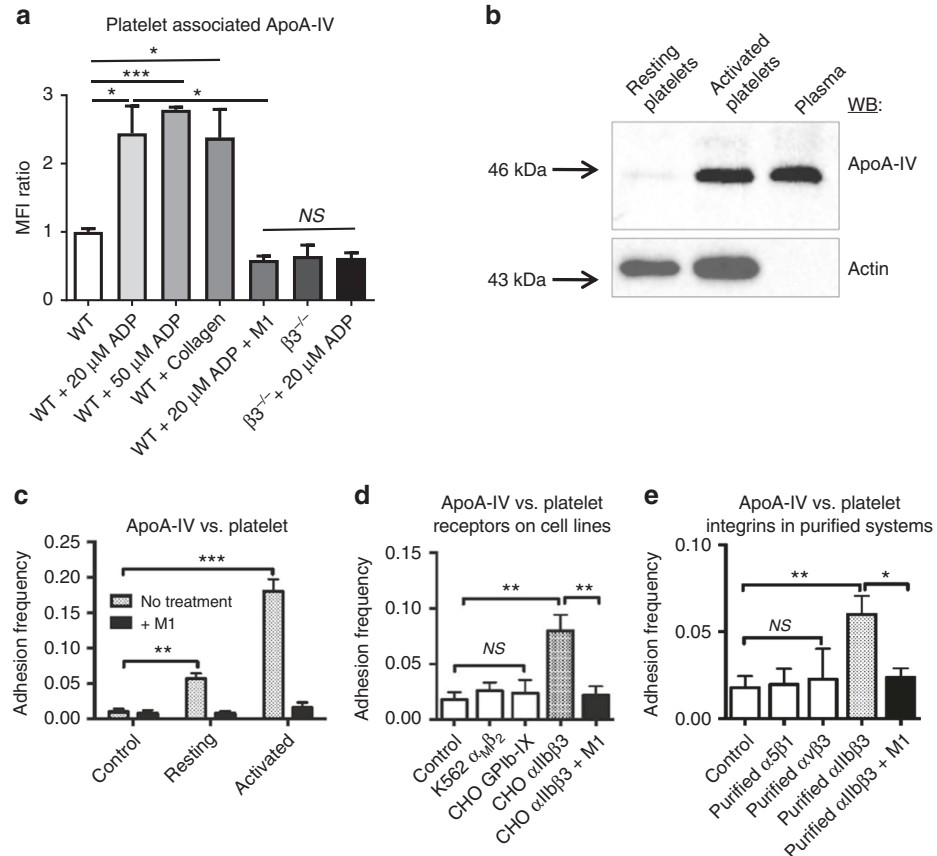

**Fig. 1** ApoA-IV is a novel ligand of αIIbβ3 integrin. **a** Flow cytometry detection of apoA-IV on activated wild-type platelets ($n = 4$). **b** Western blot detection of plasma apoA-IV on activated platelets ($n = 4$, Supplementary Fig. 16). **c–e** Biomembrane Force Probe adhesion frequency assays. ApoA-IV bound to activated platelets with high frequency and interacted minimally with resting platelets (**c**), bound to CHO cells that express αIIbβ3, but not to the cell lines that express GPIb-IX or αMβ2 integrin (**d**) and bound to purified αIIbβ3, but not αVβ3 and α5β1 integrins (**e**). Complete inhibition by mAb M1 was indicated. BSA was used as control ($n = 50$ for each bead/target pair, >3 pairs per bar or data point for **c–e**). WT wild-type, WB western blot, MFI mean fluorescence intensity, M1 a specific anti-β3 integrin monoclonal antibody, $β3^{-/-}$ β3 integrin-deficient, CHO Chinese hamster ovary. Unpaired, two-tailed Student's t-test. Mean ± SEM. NS not significant, $*P < 0.05$, $**P < 0.01$ and $***P < 0.001$

aggregation in PRP in a dose-dependent manner (80–320 μg/mL) (Fig. 3d, e). Recombinant apoA-IV also reduced platelet aggregation induced by thromboxane $A_2$ analogue (U46619), calcium ionophore (AY23187) and various doses of collagen (Supplementary Fig. 5a-b). Furthermore, apoA-IV inhibited Fg/von Willebrand Factor (VWF)-independent platelet aggregation (Supplementary Fig. 5c), which is known to be mediated by β3 integrin[27,29–31].

To further evaluate the role of endogenous apoA-IV in platelet function, we depleted human plasma apoA-IV with an anti-apoA-IV polyclonal antibody. This depletion enhanced ADP-induced platelet aggregation in healthy blood donors (Fig. 3f). Furthermore, we employed transgenic mice overexpressing mouse apoA-IV (apoA-IV-Tg), which have an approximately 3-fold increase in plasma apoA-IV compared to WT levels[35]. Consistently, apoA-IV-Tg mice had markedly attenuated platelet aggregation (Fig. 3g).

To test whether apoA-IV affects other physiological antagonist pathways, we examined platelet cAMP levels and found that they were not significantly altered by apoA-IV treatment (Supplementary Fig. 6a). ApoA-IV also did not directly affect collagen- and ADP-induced WT platelet P-selectin expression (Supplementary Fig. 6b). However, it indirectly inhibited the late phase P-selectin expression in an αIIbβ3-dependent manner, suggesting it inhibits αIIbβ3 outside-in signaling-mediated platelet activation and granule release[36] (Supplementary Fig. 6c) and apoA-IV$^{-/-}$

platelets had enhanced P-selectin expression (Supplementary Fig. 6d). Consistently, it inhibited shear-induced platelet phosphatidylserine (PS) exposure (Supplementary Fig. 7a), which is important for cell-based thrombin generation and blood coagulation (Supplementary Fig. 7b).

These in vitro studies demonstrated that apoA-IV inhibits both Fg-dependent and -independent platelet aggregation via blocking αIIbβ3-ligand interactions, which subsequently inhibits αIIbβ3 outside-in signaling-mediated platelet activation. These suggest its physiological roles in humans and animals.

**ApoA-IV aspartic acids 5/13 are required for its inhibition.** To understand the structural basis of this antiplatelet effect and localize the potential binding site in apoA-IV for platelet αIIbβ3 integrin, we first deleted both the N-terminal 38 amino acids (Δ1-38) and C-terminal 41 amino acids (Δ336–376), leaving the core helical bundle region we had identified previously[37] (residues 39–335) intact (Fig. 4a). This deletion mutant of recombinant apoA-IV (160 μg/mL) lost platelet-inhibitory function (Fig. 4b). Similar results were observed when only the N-terminus was deleted (Δ1–38, Fig. 4c, d). Interestingly, deletion of the C-terminus (Δ336–376) in the presence of an intact N-terminus enhanced the inhibitory function of apoA-IV (Fig. 4c, d). This suggests that the N-terminus plays a key role in the interaction with αIIbβ3 and the inhibition of platelet function, whereas the

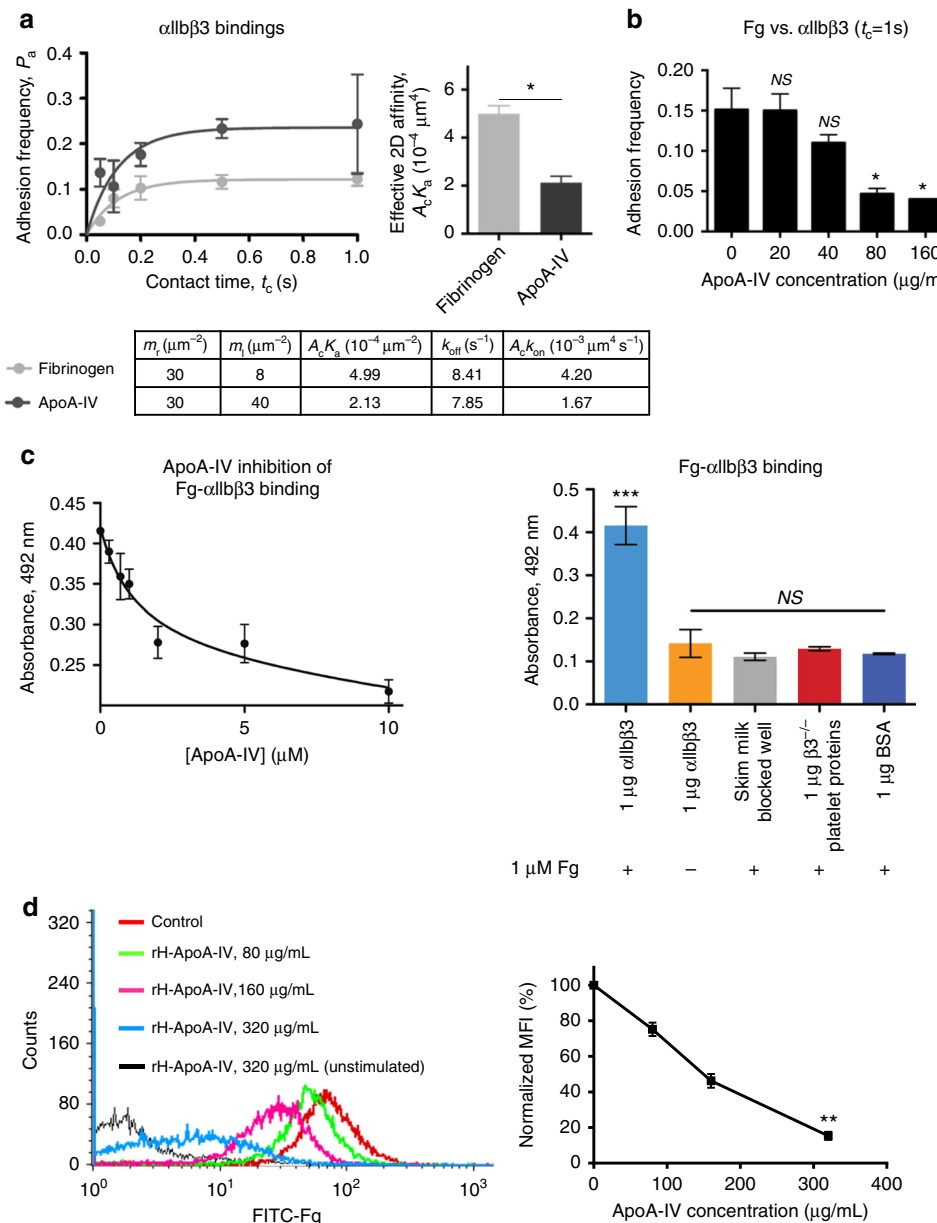

**Fig. 2** ApoA-IV competes with Fg binding to αIIbβ3 integrin. **a** Effective 2D affinity of apoA-IV-αIIbβ3 versus Fg-αIIbβ3 within contact area. The adhesion frequency ($P_a$) depends on the densities of receptor and ligand, therefore the weaker ligand apoA-IV required four-fold higher density ($40\,\mu m^{-2}$) than Fg ($8\,\mu m^{-2}$) yet only generated approximately one-fold higher $P_a$. Affinity to αIIbβ3 for apoA-IV ($2.13 \times 10^{-4}\,\mu m^4$) is 43% of that for Fg ($4.99 \times 10^{-4}\,\mu m^4$) (Grey, Fg-αIIbβ3; dark grey, apoA-IV-αIIbβ3). **b** Dose-dependent inhibition of Fg-αIIbβ3 by recombinant apoA-IV in Biomembrane Force Probe assay. **c**, **d** Dose-dependent inhibition of Fg-αIIbβ3 by recombinant human apoA-IV in ELISA (**c**) and a flow cytometry assay (**d**). The inhibitory effect of varying apoA-IV concentrations (0–10 μM) on the interaction between immobilized αIIbβ3 (1 μg) and Fg (1 μM) (**c**, left). The concentration–response curves were fit to the half maximal inhibitory model (methods) to determine the IC50 for apoA-IV. The apoA-IV IC50 for the inhibition of Fg-αIIbβ3 was found to be $1.32 \pm 0.42\,\mu M$. Control experiments confirm that the ELISA signal observed is indeed from specific binding interactions and not from non-specific or off-target binding (**c**, right). FITC-conjugated human Fg binding to the human gel-filtered platelets that were incubated with increasing doses of human recombinant apoA-IV or BSA control was analyzed by flow cytometry (**d**). Platelets were activated by TRAP (250 μM). $P_a$ adhesion frequency, $A_c$ contact area, $m_r$ αIIbβ3 density, $m_l$ Fg or apoA-IV density, $k_{off}$ 2D off-rate, $A_c k_{on}$ 2D effective on-rate, rH-ApoA-IV recombinant human apoA-IV. $n = 4$. Unpaired, two-tailed Student's $t$-test. Mean ± SEM. NS not significant, *$P < 0.05$, **$P < 0.01$ and ***$P < 0.001$

C-terminus negatively regulates this process. Our previous studies have demonstrated that the C-terminus of apoA-IV can interact with the N-terminus, which alters the lipid binding affinity of apoA-IV[38,39]. These new data suggest that this intra-molecular interaction also regulates the exposure of the apoA-IV N-terminus for binding to platelet αIIbβ3.

Notably, apoA-IV does not contain an RGD sequence, the classical binding motif of many integrin ligands. However,

through sequence analysis we observed two conserved aspartic acid (D) residues at positions 5 and 13. The D residues in some non-RGD integrin ligands have been reported to be essential for integrin binding[40]. We replaced these residues individually with glutamate (E), a conservative mutation that preserves the negative charge, but increases the side-chain volume of the amino acid. As expected, a mutation in D5E or D13E attenuated the inhibitory function of apoA-IV (Fig. 4e). We then

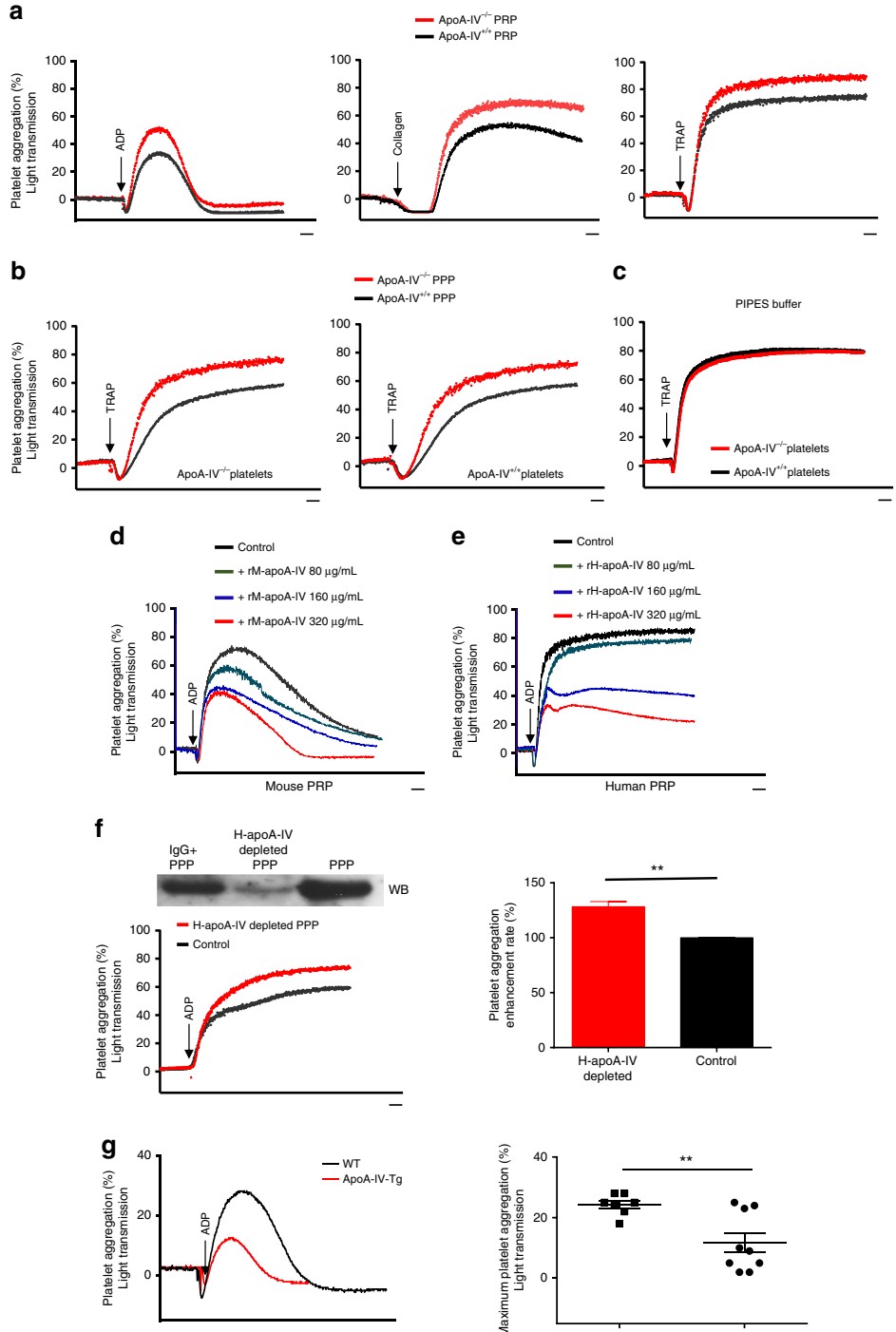

**Fig. 3** ApoA-IV inhibits platelet aggregation in vitro. **a** Enhanced platelet aggregation in platelet-rich plasma (PRP) from apoA-IV$^{-/-}$ mice ($n = 5$, $P < 0.05$) induced by ADP (1 μM), collagen (1 μg/mL) and TRAP (250 μM). **b** Enhanced apoA-IV$^{-/-}$ and apoA-IV$^{+/+}$ platelet aggregation in apoA-IV$^{-/-}$ plasma ($n = 3$, $P < 0.05$). ApoA-IV$^{-/-}$ and apoA-IV$^{+/+}$ platelet-poor plasma (PPP) were prepared and incubated with either ApoA-IV$^{-/-}$ or apoA-IV$^{+/+}$ gel-filtered platelets. Platelet aggregation was induced by TRAP (250 μM). **c** No significant difference in aggregation between gel-filtered apoA-IV$^{+/+}$ and apoA-IV$^{-/-}$ platelets in PIPES buffer induced by TRAP (250 μM) ($n = 3$). **d** Dose-dependent inhibition of ADP (1 μM) -induced platelet aggregation in mouse PRP by recombinant mouse apoA-IV. Areas under the curves for each group were compared to control ($n = 4$, $P < 0.05$). **e** Dose-dependent inhibition of ADP (2.5 μM) -induced platelet aggregation in human PRP by recombinant human apoA-IV ($n = 4$, $P < 0.05$). **f** Depletion of human plasma apoA-IV from healthy donors with different ethnic backgrounds, by a goat anti-human apoA-IV IgG, enhanced ADP-induced platelet aggregation ($n = 4$). Non-specific goat IgG treated plasma was used as control. ApoA-IV-depleted PPP was shown by western blot (Supplementary Fig. 17). **g** Transgenic mice overexpressing mouse apoA-IV had markedly attenuated platelet aggregation induced by ADP (1 μM) ($n = 4$). Aggregation traces are representative of several independent experiments indicated. The maximum platelet aggregation in each group was compared to control unless otherwise indicated. *PPP* platelet poor plasma, *rM-apoA-IV* recombinant mouse apoA-IV, *rH-apoA-IV* recombinant human apoA-IV, *ApoA-IV-Tg* transgenic mice overexpressing mouse apoA-IV. Unpaired, two-tailed Student's t-test or non-parametric Kruskal-Wallis one-way analysis of variance for multiple paired comparisons. Mean ± SEM. **$P < 0.01$. Scale bars: 2 minute (**a**–**g**)

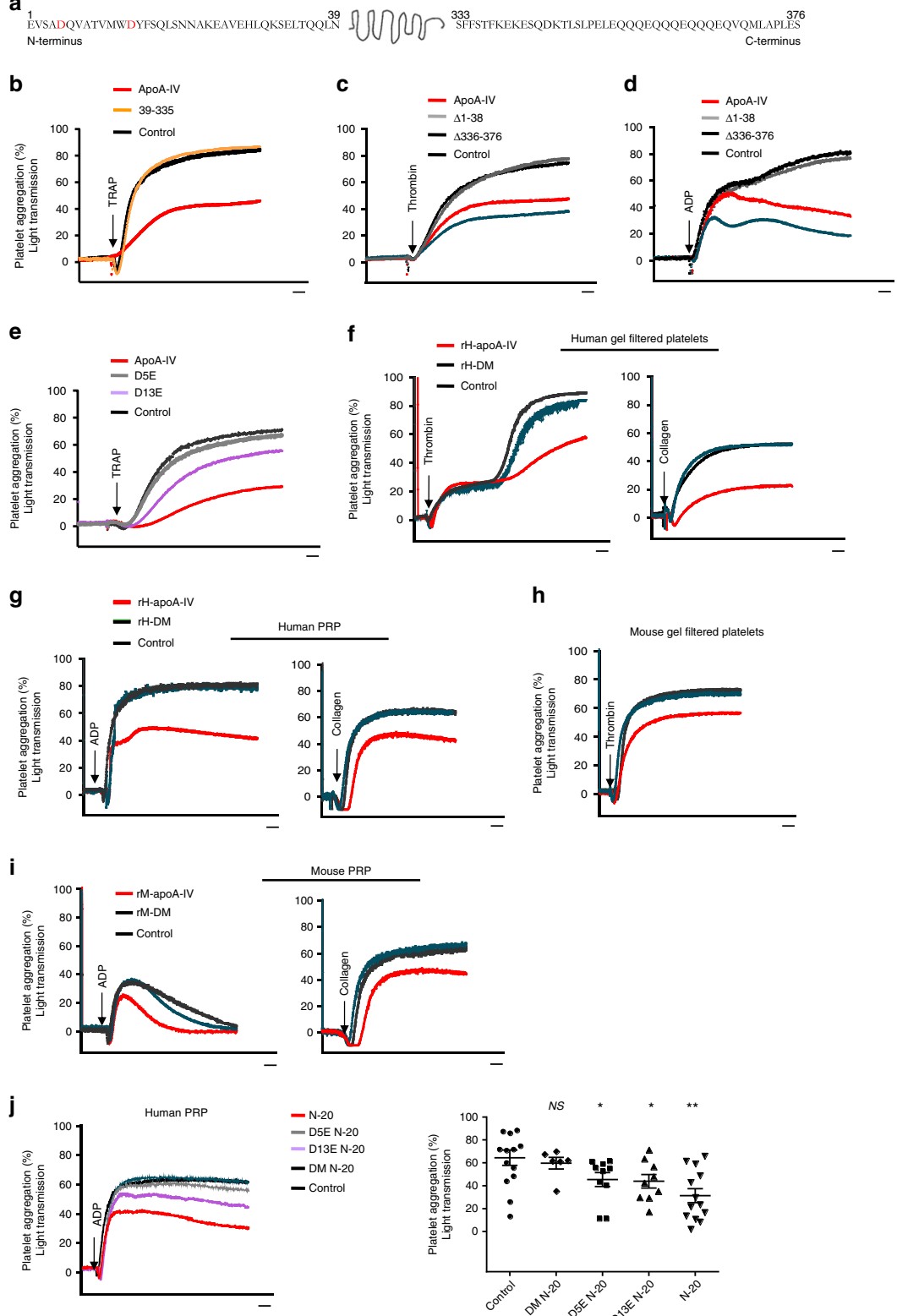

generated recombinant human and mouse double D mutant (DM) apoA-IV with mutations at both of these residues and found that DM apoA-IV abrogated these inhibitory effects (Fig. 4f, i). Consistently, BFP adhesion frequency assay showed that DM apoA-IV did not significantly bind to αIIbβ3 (Supplementary Fig. 8), resulting in the loss of inhibitory function of apoA-IV (Fig. 4f, i).

To determine whether the double D mutations affected the structure of apoA-IV, we performed circular dichroism (CD) spectroscopy and thermal denaturation assays. CD wavelength scans revealed excellent superimposition between the WT and DM apoA-IV CD spectra, suggesting no loss of secondary structural elements as a result of the mutation (Supplementary Fig. 9a). In addition, both the WT and DM apoA-IV exhibited

**Fig. 4** N-terminal aspartic acids (D5 and D13) of apoA-IV are required for its inhibitory effect on platelets. **a** Schematic map of human apoA-IV. **b** Deletion of both the N-terminus and C-terminus of human apoA-IV abolished the inhibition of platelet aggregation induced by TRAP (250 μM) (39–335 vs Control: NS; ApoA-IV vs Control: $P < 0.01$). **c, d** Recombinant apoA-IV lacking the N-terminus (Δ1–38) abolished the inhibition of platelet aggregation (Δ1–38 vs Control: NS), while only deletion of the C-terminus (Δ336–376) enhanced the inhibition induced by (**c**) thrombin (0.5 U) and (**d**) ADP (2.5 μM) (Δ336–376 vs ApoA-IV: $P < 0.05$). **e** Point mutation of aspartic acids (D5E or D13E) at the apoA-IV N-terminal attenuated the inhibition of platelet aggregation (D5E and D13E vs ApoA-IV: $P < 0.05$). **f–i** Double D (D5 and D13) mutant human apoA-IV (**f, g**) and mouse apoA-IV (**h, i**) abrogated the inhibition of platelet aggregation (DM vs Control: NS; ApoA-IV vs Control: $P < 0.01$). Area under the curve of each group was compared. **j** Synthetic peptides of N-terminal 20 amino acids (N-20) of apoA-IV inhibited human platelet aggregation, while D5E, D13E or DM mutations on the N-20 synthetic peptides (D5E N-20, D13E N-20 and DM N-20, respectively) attenuated or abrogated its inhibition. For each group, the maximum platelet aggregation was compared to control. Aggregation traces are representative of four independent experiments. *rH-DM* double D mutant human apoA-IV, *rM-DM* double D mutant mouse apoA-IV. $n = 4$. Unpaired, two-tailed Student's t-test or non-parametric Kruskal-Wallis one-way analysis of variance for multiple paired comparisons. Mean ± SEM. NS not significant, *$P < 0.05$ and **$P < 0.01$. Scale bars: 2 minute (**b-j**)

---

similar thermal denaturation profiles with a melting temperature ($T_m$) of 54.3 °C (Supplementary Fig. 9b). Taken together, these findings strongly suggest that double D mutations at the apoA-IV N-terminus do not affect the global structure and stability of apoA-IV. Furthermore, the synthetic peptide of the N-terminal 20 amino acids was capable of inhibiting platelet aggregation, while the D5E, D13E, or DM mutations on the peptides attenuated or abrogated its inhibition (Fig. 4j). These studies demonstrate that D5 and D13 at the apoA-IV N-terminus are required for its inhibitory function and are potential binding sites for αIIbβ3 integrin.

**ApoA-IV inhibits thrombus growth in vitro**. To examine whether apoA-IV affects thrombus formation under flow conditions, we perfused heparin-anticoagulated and fluorescently labeled whole blood from either apoA-IV$^{-/-}$ or apoA-IV$^{+/+}$ mice across type I collagen-coated microcapillary perfusion chambers, under a real-time fluorescence microscope. Compared to apoA-IV$^{+/+}$ controls, platelet aggregation and thrombus growth were enhanced in apoA-IV$^{-/-}$ mouse blood both at 300 s$^{-1}$ (Fig. 5a, Supplementary Fig. 10a) and 1800 s$^{-1}$ (Fig. 5b, Supplementary Fig. 10b) wall shear rates, corresponding to blood flow in venules/large arteries and arterioles, respectively. Consistently, addition of recombinant mouse apoA-IV (160 μg/mL) to heparin-anticoagulated WT mouse blood significantly attenuated the thrombus growth under both low (Fig. 5c) and high (Fig. 5d) shear rates, while DM apoA-IV did not (Fig. 5c, d). Recombinant mouse apoA-IV also markedly inhibited mouse thrombus growth under shear rates that mimic stenotic arteries (e.g., 5000 s$^{-1}$; Supplementary Fig. 10c). Human thrombus growth under both low (Fig. 6a) and high (Fig. 6b) shear rates in the perfusion chambers was also inhibited by recombinant human apoA-IV. These ex vivo results suggest that apoA-IV is a significant inhibitor of thrombosis at both arterial and venous shear rates.

Interestingly, we observed that the inhibitory effect of apoA-IV increased with increasing shear rate (Figs. 5c, d and 6, Supplementary Fig. 10–11). Considering the results obtained with C-terminal truncated apoA-IV in platelet aggregation (Fig. 4c, d), it is possible that high shear stress may disrupt the previously reported intra-molecular contact between the C and N termini[38], increasing the exposure of N-terminal residues that bind to platelet αIIbβ3. The enhanced platelet inhibition with increasing shear rate may be a critical property of apoA-IV opposing the formation of occluding plugs at the apex of growing thrombi, thus preserving blood supply to the downstream tissues.

**ApoA-IV inhibits thrombus growth in vivo**. To examine whether apoA-IV affects thrombus growth in vivo, we next utilized

two complementary intravital microscopy thrombosis models and a large artery thrombosis model. Thrombus growth and vessel occlusion induced by FeCl₃ injury[29–31,41] were significantly accelerated in apoA-IV$^{-/-}$ mice versus apoA-IV$^{+/+}$ littermate controls (Fig. 7a). To exclude the possibility that apoA-IV in vivo inhibition was due to anti-oxidant activity[42] as opposed to blockade of platelet αIIbβ3 integrins, we used a cremaster arteriole intravital microscopy thrombosis model which does not involve oxidative stress as it induces mild vascular injury with a laser[43,44]. We found that thrombus growth was also markedly accelerated and thrombus dissolution was slower in apoA-IV$^{-/-}$ mice than apoA-IV$^{+/+}$ controls (Fig. 7b). Interestingly, we observed that thrombi could reach occlusive size in apoA-IV$^{-/-}$ mice (Fig. 7b), which has not been previously reported in this laser injury model[43]. These findings indicate that endogenous apoA-IV could inhibit thrombus growth and promote thrombus dissolution. Intravenous infusion (9 μg/g) of recombinant mouse apoA-IV attenuated the thrombus growth in apoA-IV$^{-/-}$ mice (Fig. 7b). To examine whether apoA-IV also inhibits thrombus growth in large arteries, a carotid artery thrombosis model[45,46] was used. ApoA-IV$^{-/-}$ mice indeed exhibited accelerated vessel occlusion in carotid arteries (Fig. 7c). We further demonstrated that thrombus growth in WT mice was significantly inhibited by intravenous infusion of recombinant apoA-IV, but not DM apoA-IV, in mesenteric arterioles (Fig. 8a), cremaster arterioles, and carotid arteries Fig. 8b). ApoA-IV significantly prevented stable vessel occlusion in carotid arteries, suggesting its critical inhibitory roles in high shear flow environments (Fig. 8b). These data clearly demonstrate that apoA-IV is an endogenous antithrombotic factor.

To further evaluate the therapeutic potential of recombinant apoA-IV, we compared apoA-IV treatment (9 μg/g) with aspirin (20 μg/g) and clopidogrel (12 μg/g) treatments. The inhibitory effect of apoA-IV was similar to aspirin or clopidogrel treatments, although there was a synergistic effect when apoA-IV and aspirin, or apoA-IV and clopidogrel were used in conjunction (Supplementary Fig. 12). Importantly, there was no significant alteration of bleeding time or blood loss between apoA-IV$^{-/-}$ mice and littermate controls, or in mice transfused with apoA-IV (Supplementary Fig. 13). No bleeding tendency has been observed during surgery on these mice in the intravital microscopy experiments after apoA-IV infusion (9 μg/g), although infusing a higher dose (>45 μg/g, i.e., >5 times of physiological level) of recombinant apoA-IV prolonged the bleeding time. These data are consistent with the observations in heterozygous β3 integrin deficient (β3$^{+/-}$) mice that express approximately 50% αIIbβ3 on platelets without obvious bleeding disorders but do have impaired thrombosis, particularly at high shear in vitro and in vivo (Supplementary Fig. 14a-d). In addition, plasma lipid and cholesterol components in apoA-IV transfused mice were not significantly altered (Supplementary Table 1). Therefore,

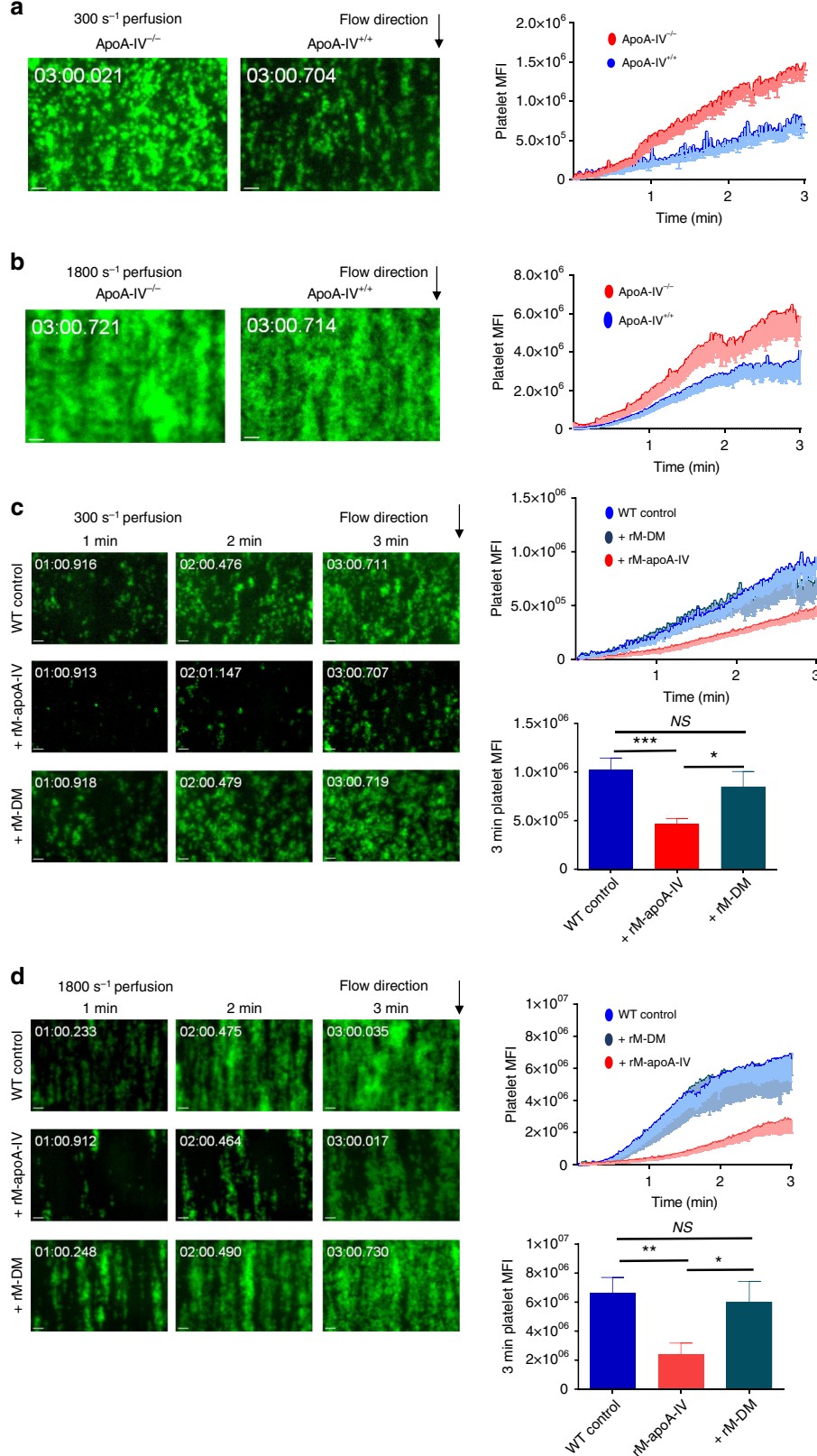

**Fig. 5** ApoA-IV inhibits mouse thrombus growth in vitro. **a** Enhanced thrombus formation in blood from apoA-IV$^{-/-}$ mice under low shear rates. **b** Enhanced thrombus formation in blood from apoA-IV$^{-/-}$ mice under high shear rates. **c**, **d** Inhibition of thrombus formation in blood from WT mice by recombinant mouse apoA-IV under low shear (**c**) and high shear (**d**), while double D mutations abolished its inhibition. Representative images of thrombus formation (green) in four individual experiments were shown. $n = 9$–12 thrombi. Unpaired, two-tailed Student's $t$-test or non-parametric Kruskal–Wallis one-way analysis of variance for multiple paired comparisons. Mean ± SEM. NS not significant, *$P < 0.05$, **$P < 0.01$ and ***$P < 0.001$. Scale bars: 10 μm (**a**–**d**)

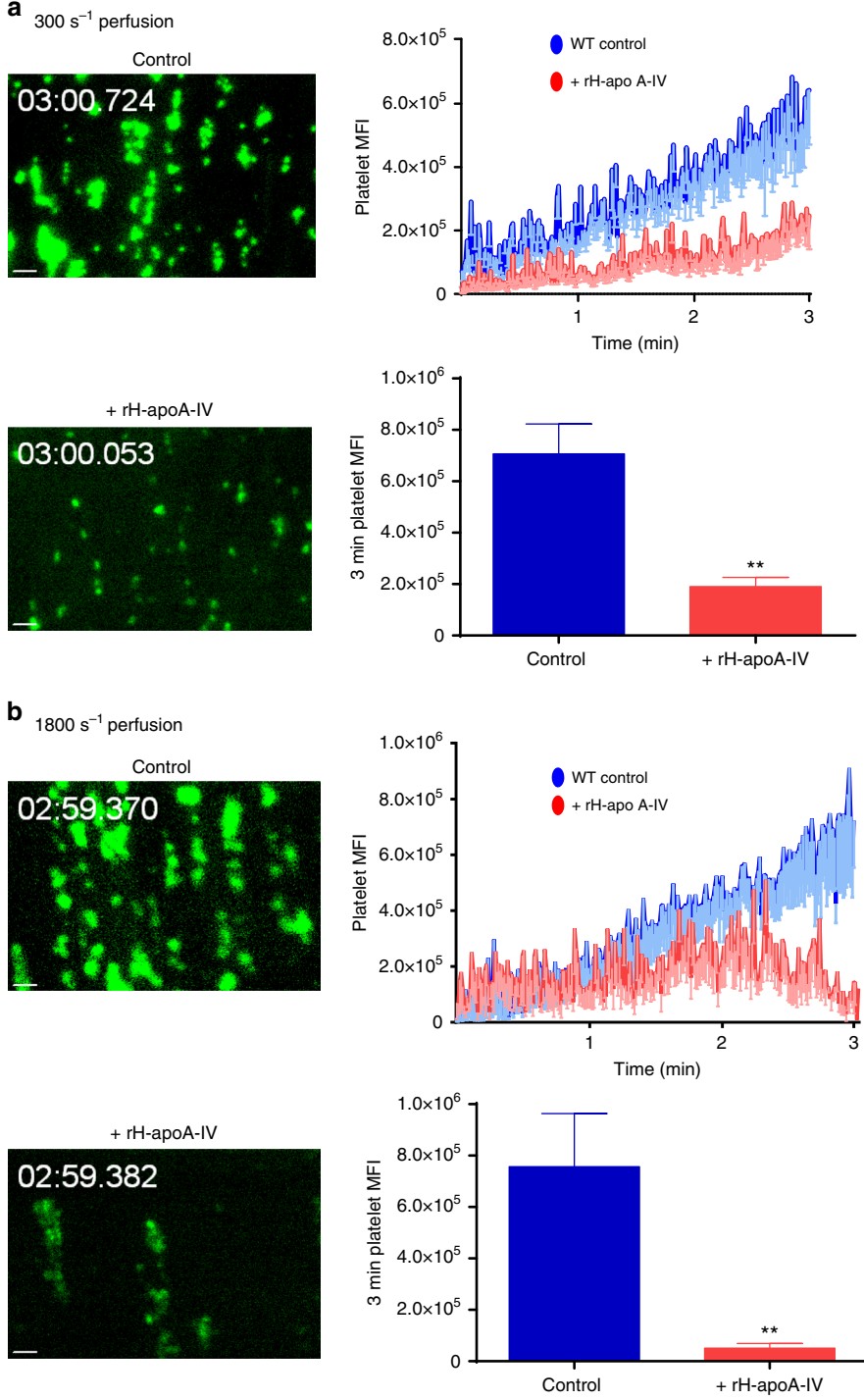

**Fig. 6** ApoA-IV inhibits human thrombus growth in vitro. Recombinant human apoA-IV inhibited thrombus formation in human blood under both low (**a**) and high (**b**) shear rates. Representative images of thrombus formation (green) in four individual experiments were shown. $n = 9$ thrombi. Unpaired, two-tailed Student's $t$-test. Mean ± SEM. **$P < 0.01$. Scale bars: 10 μm (**a**, **b**)

apoA-IV may have the potential for treating thrombotic diseases with a relatively safe therapeutic window.

**ApoA-IV attenuates postprandial platelet hyperactivity.** ApoA-IV can be rapidly synthesized and secreted in response to lipid intake[2,47]. To investigate the in vivo role of apoA-IV following high fat diet (HFD), we tested platelet function from C57BL/6 J WT, apoA-IV-Tg, apoA-IV$^{+/+}$, heterozygous (apoA-IV$^{+/-}$),

and apoA-IV$^{-/-}$ mice under fasting conditions, and after a HFD. We observed that a HFD induced postprandial platelet hyperactivity in WT mice (Fig. 9a). Transgenic mice overexpressing mouse apoA-IV, with an increase in plasma apoA-IV concentration following HFD, markedly attenuated postprandial platelet aggregation (Fig. 9a). Correspondingly, deficiency or decrease of apoA-IV levels significantly enhanced postprandial platelet function (Fig. 9b). Thus, apoA-IV is a postprandial regulator of platelet activity in vivo.

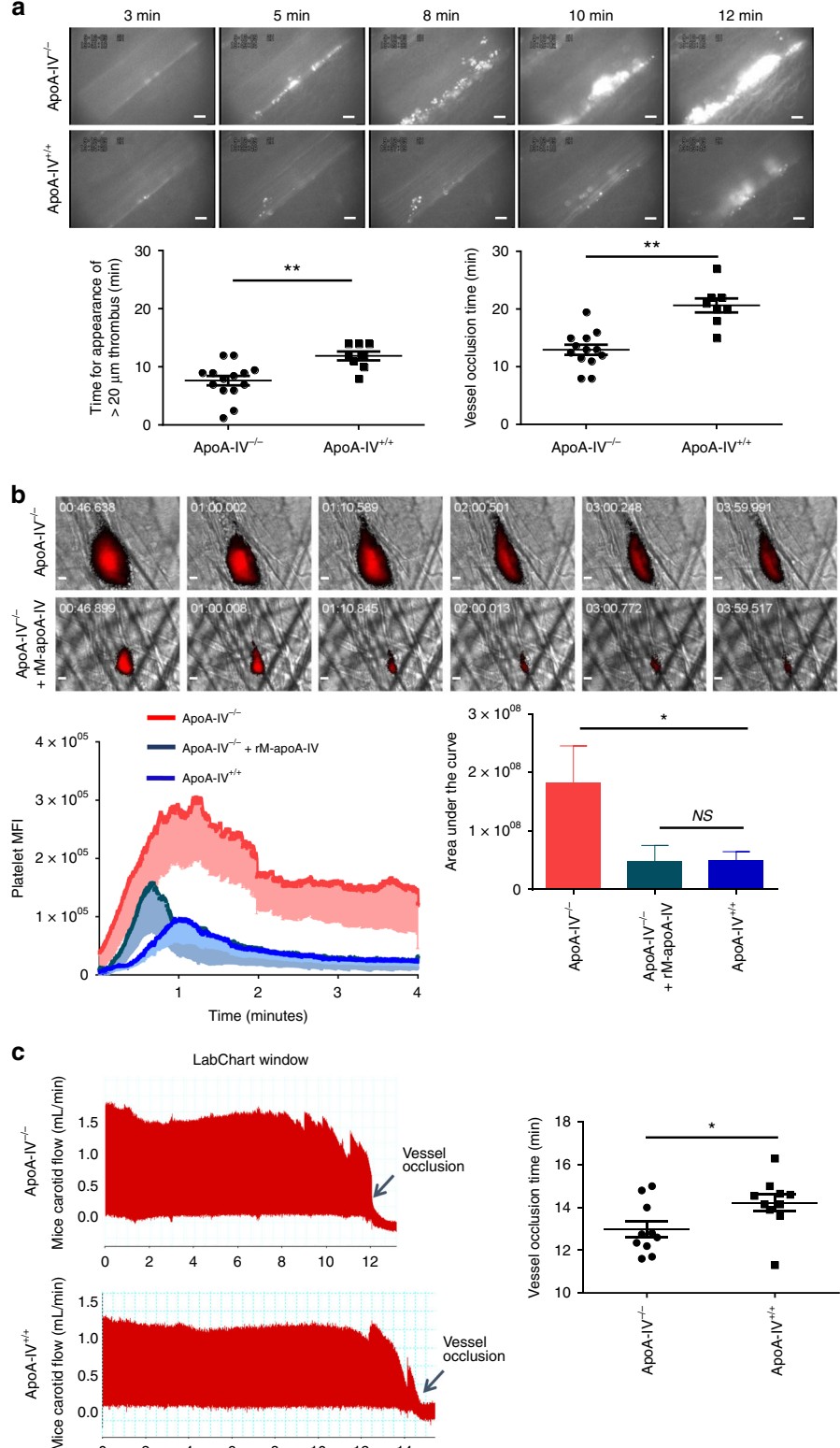

**Fig. 7** ApoA-IV deficiency results in enhanced thrombus growth and vessel occlusion in vivo. **a** Accelerated thrombus growth (white) in apoA-IV$^{-/-}$ mice in a FeCl$_3$ (4%) injury mesenteric arterioles model ($n = 14$). **b** Accelerated thrombus growth and occlusive thrombi (red) in apoA-IV$^{-/-}$ mice in a laser injury cremaster arteriole model. Infusion of recombinant mouse apoA-IV decreased the thrombus formation ($n = 23$ thrombi in 4 mice). **c** Accelerated vessel occlusion in apoA-IV$^{-/-}$ mice in a FeCl$_3$ (10%) injury carotid artery model ($n = 10$). Representative images of thrombus growth in all mice tested were shown. Unpaired, two-tailed Student's $t$-test. Mean ± SEM. NS not significant, *$P < 0.05$ and **$P < 0.01$. Scale bars: 10 μm (**a**, **b**)

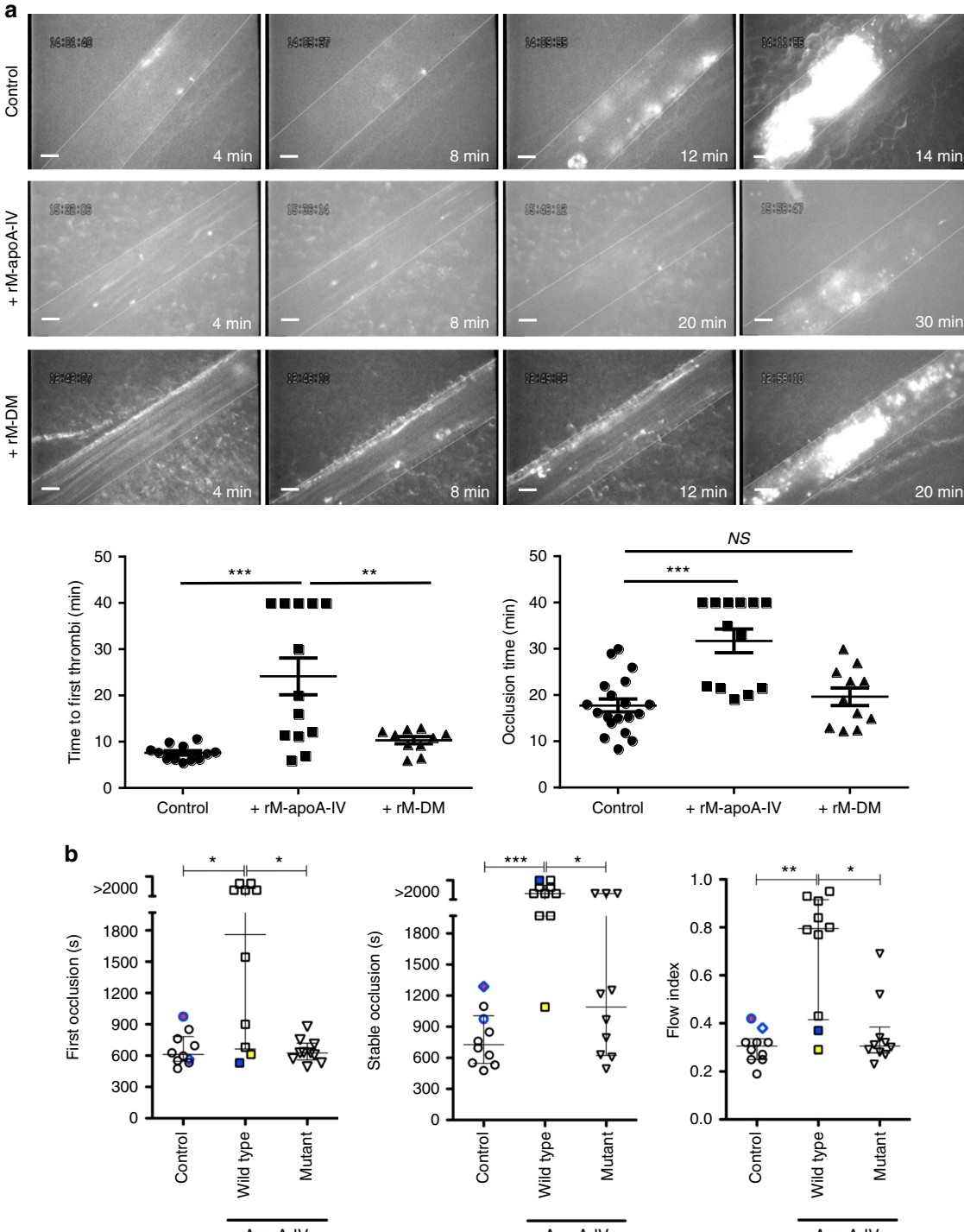

**Fig. 8** Intravenous infusion of recombinant apoA-IV inhibits thrombus growth and vessel occlusion in vivo. **a** Thrombus growth (white) in wild-type mice mesenteric arterioles was inhibited by infusion of recombinant mouse apoA-IV (rM-apoA-IV; $n = 13$) but not double D mutant apoA-IV (rM-DM; $n = 11$). Representative images of thrombus growth in all mice tested is shown. **b** ApoA-IV prevented stable occlusion in a mouse FeCl$_3$ injury carotid artery thrombosis model. The time to first (left) and stable (middle) occlusion was significantly prolonged and the flow index (right) increased (indicating less decrease in blood flow) by infusion of rM-apoA-IV as compared to vehicle (control) or rM-DM ($n = 10$). Individual experimental points are shown along with median and interquartile range. In the control group, a circle and diamond with thick blue borders identify two cases that gave a value outside of the 90th percentile (red filling) for at least one of the parameters measured; note that abnormalities in the measured parameters are not concordant. In the apoA-IV WT treatment (i.e., rM-apoA-IV) group, a blue and a yellow square identify the two cases in which treatment caused the least change in the blood flow index, highlighting that one of them failed to achieve stable occlusion; thus, delayed stable occlusion is the most sensitive parameter to discriminate the antithrombotic effect of rM-apoA-IV infusion. All significantly different comparisons are shown. *apoA-IV WT* recombinant mouse apoA-IV, *apoA-IV mutant* double D mutant apoA-IV. Non-parametric Kruskal–Wallis one-way analysis of variance for multiple paired comparisons. Mean ± SEM. NS not significant, *$P < 0.05$, **$P < 0.01$ and ***$P < 0.001$. Scale bars: 10 μm (**a**)

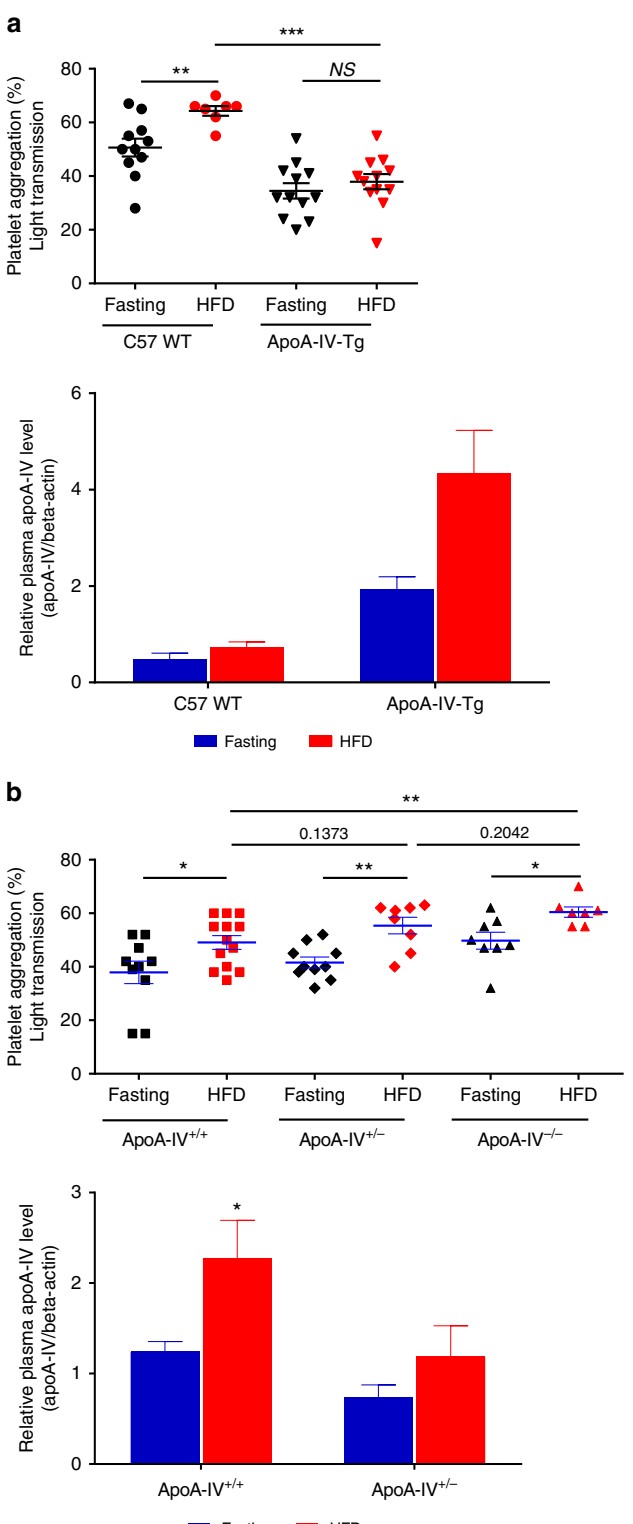

**a**

**b**

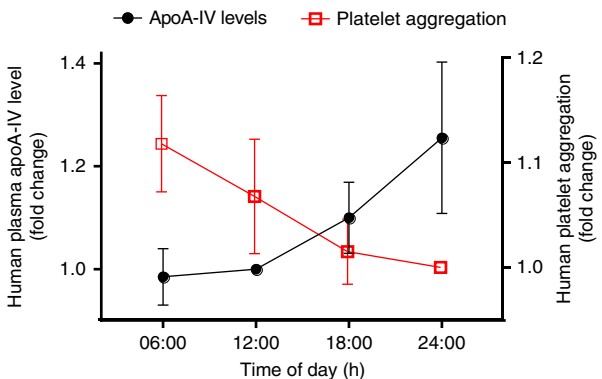

**Fig. 9** ApoA-IV attenuates postprandial platelet hyperactivity. **a** Transgenic mice overexpressing apoA-IV attenuated high fat diet (HFD)-induced postprandial platelet hyper-reactivity. C57 WT mice and apoA-IV transgenic mice (apoA-IV-Tg) were fasted for 12 h and followed by HFD for 3 h. WT mice showed a significant enhancement in platelet aggregation after acute HFD. ApoA-IV-Tg mice showed a prevention of postprandial platelet hyper-reactivity (top). HFD increased the plasma levels of apoA-IV in apoA-IV-Tg mice (bottom). **b** Deficiency of apoA-IV markedly increased postprandial platelet function (top, apoA-IV$^{-/-}$ vs apoA-IV$^{+/+}$, $P < 0.01$). HFD increased the plasma levels of apoA-IV in apoA-IV$^{+/+}$ and apoA-IV$^{+/-}$ mice (bottom). *HFD* high fat diet. $n = 12$ mice/group from three individual experiments. Unpaired, two-tailed Student's $t$-test or non-parametric Kruskal–Wallis one-way analysis of variance for multiple paired comparisons. Mean ± SEM. NS not significant, *$P < 0.05$, **$P < 0.01$ and ***$P < 0.001$

**Fig. 10** ApoA-IV concentrations and platelet aggregation levels throughout the day are predominantly inversely correlated in humans. Humans have the strongest platelet aggregation in the morning at 06:00 (Red: 06:00 vs 18:00, $P < 0.01$; 06:00 vs 24:00, $P < 0.05$). Plasma apoA-IV levels in humans exhibited a midnight peak and nadir in the morning at 06:00 (Black: 06:00 vs 18:00 and 24:00, $P < 0.05$). Data are normalized according to platelet aggregation levels at 24:00 and apoA-IV levels at 12:00. $n = 20$ donors. Unpaired, two-tailed Student's $t$-test. Mean ± SEM

strongest platelet aggregation in the morning around 06:00 (Fig. 10, Supplementary Fig. 15a). Interestingly, plasma apoA-IV levels in humans also exhibited a circadian rhythm, with a peak at midnight and nadir around 06:00 (Fig. 10, Supplementary Fig. 15b). Importantly, our study revealed that the levels of apoA-IV and platelet aggregation are, in most cases (16/20), inversely correlated throughout the day (Fig. 10, Supplementary Fig. 15a-d). Therefore, the lowest level of apoA-IV in the morning likely contributes to enhanced platelet aggregation, leading to more adverse cardiovascular and cerebrovascular events.

## Discussion

Thrombotic disorders, such as heart attack and stroke, are the leading causes of mortality and morbidity worldwide. Platelet aggregation mediated by αIIbβ3 integrin and its ligands is essential for the progression of these disorders. In this comprehensive study, using a series of in vitro and in vivo methods (including the single-molecule force spectroscopy technique BFP), as well as conducting human studies in healthy subjects, we identified that apoA-IV is a novel ligand of platelet αIIbβ3 integrin and an endogenous inhibitor of platelet aggregation and thrombosis. Through the long-term moderate but significant attenuation of platelet hyperactivity, particularly during the postprandial period, apoA-IV may also slow down the chronic

**Human apoA-IV inversely correlates with platelet aggregation**. Serious life-threatening cardiovascular events peak in the morning, likely related to the increased thrombosis in critical vessels[48–50]. Besides changes in apoA-IV postprandial concentrations, whether there are diurnal changes in apoA-IV levels in humans that might contribute to the increased incidence of heart attacks in the morning is unknown. We therefore studied the relationship between levels of apoA-IV and platelet aggregation at 6-h intervals for 24 h in 20 healthy volunteers. We found that humans exhibit the

process of atherosclerosis and other inflammatory diseases[7,35,51] without impairing hemostasis.

Our results demonstrate that endogenous apoA-IV can block 20–40% of interactions between αIIbβ3 and Fg or other pro-thrombotic ligands under static or low shear rate conditions. Notably, clinical experience has not revealed pronounced bleeding effects of a 50% reduction in αIIbβ3 expression (heterozygous individuals who are carriers for Glanzmann's thrombasthenia). The same holds true in β3[+/−] mice. However, thrombosis is different from hemostasis and the antithrombotic effect could be achieved before bleeding occurs, as has been well documented in both animal models and antithrombotic drug developments in human clinical trials[52–55]. Such is the case in our thrombosis models, in which thrombus growth was markedly decreased in β3[+/−] mice in vitro and in vivo. Since there are limited human cases of Glanzmann's thrombasthenia, it is currently unknown whether the 50% reduction in αIIbβ3 can also reduce their risk for thrombosis. This concept is consistent with the positive correlation between levels of Fg (αIIbβ3 ligand) and platelet aggregation in vitro, and thrombotic events in patients in vivo[56,57].

Of note, apoA-IV has an enhanced antiplatelet and antithrombotic effect with escalating shear stress at the apical zone of growing thrombi under flow conditions, likely due to high shear stress that may disrupt apoA-IV intra-molecular interactions and increase exposure of the apoA-IV binding site for αIIbβ3. This may be critical for preventing vessel occlusion at the sites of stenosis. In addition, competitive blockade of the Fg-αIIbβ3 interaction and αIIbβ3 outside-in signaling can subsequently reduce platelet granule release and PS exposure, which may decrease amplification of platelet aggregation, and thrombin generation/clot formation in non-anticoagulated blood[58,59]. These mechanisms may explain its stronger antithrombotic effects in vivo than in vitro. Thus the blockade of αIIbβ3 by apoA-IV is biologically significant in that this level of inhibition does not significantly impair hemostasis, but can predominantly attenuate thrombosis, while a low level of plasma apoA-IV increases the risk for CVDs[8,9]. Indeed, our finding that circadian rhythm of apoA-IV in most blood donors had a negative correlation with platelet aggregation throughout a 24 h period, is suggestive and might at least partially explain the high incidence of heart attack and stroke in the morning compared to the evening.

Recent studies have shown that the high density lipoprotein (HDL) inhibits platelet function by promoting cholesterol efflux, as well as by attenuating agonist-induced platelet activation signaling pathways upon binding to platelet HDL receptors[60–62]. Although a small fraction of apoA-IV in circulation is associated with HDL particles, the inhibitory effects observed in our in vitro study do not likely result from other HDL components, but rather via apoA-IV directly antagonizing prothrombotic ligand-αIIbβ3 binding. Recombinant apoA-IV in buffer (i.e., in the absence of HDL components) directly bound to αIIbβ3 in single molecule BFP assays and inhibited gel-filtered platelet aggregation. Double mutations of apoA-IV D5E and D13E, which do not affect global structure and likely do not affect affinity of lipid binding[39], completely abolished the binding to αIIbβ3 and abrogated the inhibitory effects on platelet function in vitro and in vivo. Furthermore, apoA-IV did not significantly alter platelet cAMP levels. However, the current study could not completely exclude other mechanisms mediated by apoA-IV, such as possible interactions with HDL, LDL or membrane lipid components[25,62], which may synergistically contribute to the inhibitory effect on platelet aggregation and thrombus growth. It is also conceivable that interactions with HDL and other plasma components may increase the sensitivity of apoA-IV to shear stress, which facilitate the accessibility of the N-terminus to αIIbβ3 integrin. These possibilities merit future investigation.

The displacement of platelet β3 integrin ligands has broad implications not only in thrombotic disorders, but also in platelet-mediated inflammation. Platelet β3 integrins contribute to platelet–leukocyte and platelet–endothelial cell interactions, and the effect of β3 integrin inhibitors on reduction of intimal hyperplasia has been well demonstrated in animal models[63]. We previously reported that platelet Fg-αIIbβ3 integrin interaction can induce platelets to de novo synthesize P-selectin[64], an important mediator of inflammation, leukocyte rolling and the Th1 immune response, which promotes atherogenesis[17,21,65]. In the current study, although we did not observe that recombinant apoA-IV directly altered platelet activation and P-selectin expression, apoA-IV indirectly inhibited the late phase P-selectin expression by interfering with αIIbβ3-Fg/prothrombotic ligand interactions and the subsequent αIIbβ3 outside-in signaling, which is important for platelet granule release and P-selectin expression[36]. Notably, in the absence of αIIbβ3, apoA-IV did not significantly inhibit the late phase P-selectin expression on thrombin-treated β3[−/−] platelets. Consistently, enhanced P-selectin expression was observed on ADP-treated platelets from apoA-IV[−/−] PRP, where Fg and other prothrombotic ligands can more easily bind to αIIbβ3 and deliver outside-in signaling, which may increase both platelet P-selectin synthesis and releasing. Our data may explain why transgenic mice overexpressing apoA-IV were protected from atherosclerosis[51].

It has also been reported that postprandial spikes in glucose and lipids may generate excess free radicals, induce inflammation and enhance platelet-endothelial cell and platelet-leukocyte interactions, which may play an important role in atherogenesis and thrombogenesis[25,66]. In our study, we also observed that a HFD induced postprandial platelet hyperactivity. The rapid increase in apoA-IV secretion after lipid intake is therefore important to attenuate postprandial platelet hyperactivity and inflammation, and thus may be physiologically critical to slow the progression of atherosclerosis and thrombogenesis. Furthermore, since platelets have recently been revealed to contribute to tumor metastasis[23], in which αIIbβ3-mediated platelet-tumor cell heterotypic aggregation is involved[23], whether apoA-IV can thus attenuate this process remains to be investigated.

This study provides the first direct link between apoA-IV and platelet activity, and the novel mechanisms as to why apoA-IV is a protective factor for CVDs. We demonstrate that apoA-IV, by antagonizing platelet αIIbβ3 integrin activity, attenuates platelet aggregation and thrombosis, postprandial platelet hyperactivity, and may also slow the progression of atherosclerosis (i.e., both early and late stages of atherothrombosis). Furthermore, we identified a circadian rhythm of apoA-IV in humans, which negatively correlates with platelet aggregation and the risk of cardiovascular events. Control of food consumption (e.g., increase of unsaturated fat)[2], which raises apoA-IV synthesis and secretion, may have a significant impact on the prevention of the early developmental stages of atherosclerosis. Infusion of recombinant apoA-IV may be able to directly intervene in thrombosis and control CVDs and stroke.

## Methods

**Materials**. Thrombin, ADP, streptavidin and human fibrinogen (Fg) were purchased from Sigma-Aldrich (Oakville, ON, Canada). Thrombin receptor activating peptide (TRAP; AYPGKF-NH₂) was purchased from Peptides International (Louisville, MO, USA). Type I collagen fibrils (equine collagen Horm) were purchased from Nycomed (Roskilde, Denmark). Alexa Fluor 488 conjugate, DiOC6 dye, and Calcein AM were purchased from Invitrogen (Burlington, ON, Canada). Goat anti-human apoA-IV antibody (N-20, sc-19036; C-20, sc-19038), mouse anti-human apoA-IV antibody (G-8; sc-374543), mouse IgG kappa binding protein conjugated to HRP (sc-516102), fluorescein isothiocyanate (FITC)-conjugated

donkey anti-goat IgG, and FITC-conjugated human Fg were purchased from Santa Cruz (Dallas, Texas, USA). Rabbit anti-goat IgG HRP conjugated antibody (HAF017) was purchased from Bio-Techne (Minneapolis, MN, USA). Rat anti-mouse CD41 antibody was purchased from EMFRET Analytics (Eibelstadt, Germany). Maleimide-PEG2-Biotin was purchased from Thermo Scientific (Canada). Quick Change Site-Directed Mutagenesis Kit was purchased from Agilent Technologies (CA, USA). Immobilized metal affinity chromatography columns (His GraviTrap) were purchased from GE Healthcare (Piscataway, NJ, USA). Purified human platelet β3 integrins were purchased from Molecular Innovations (Novi, MI, USA).The high fat food (TD.88137) was purchased from the Harlan Laboratories (Madison, WI, USA).

Recombinant α5β1-Fc ectodomain was a gift from Dr. Martin J. Humphries (University of Manchester, UK). Recombinant αVβ3 ectodomain with a hexa-Histidine tag at the C-terminus was a gift from Dr. Junichi Takagi (Osaka University, Japan). K562 cell line expressing αMβ2 was provided by Dr. Tanya Mayadas (Harvard University, USA). CHO cell lines expressing GPIb-IX complex and αIIbβ3 were gifts from Dr. Larry McIntire (Georgia Institute of Technology, USA) and Dr. Xiaoping Du (University of Illinois, USA) respectively. Cell lines were tested for mycoplasma contamination before experiments.

**Mice.** ApoA-IV deficient (apoA-IV$^{-/-}$) mice and C57BL/6J wild-type (WT) mice were purchased from Jackson Laboratory (Bar Harbor, USA). ApoA-IV$^{-/-}$ mice have been backcrossed onto a C57BL/6J background to generate apoA-IV$^{+/+}$ littermates. ApoA-IV transgenic mice overexpressing mouse apoA-IV (apoA-IV-Tg) were kindly provided by Dr. Karen Reue (University of California, USA). β3 integrin gene-deficient (β3$^{-/-}$) mice were kindly provided by Dr. Richard O. Hynes (Massachusetts Institute of Technology, USA). Both female and male mice were used with a ratio approximately 1:1 unless where indicated for the intravital microscopy thrombosis models that only male mice can be used[29,43]. Genotypes of all experimental animals were confirmed by polymerase chain reaction (PCR) analysis of tissue DNA. All mice were housed in the research vivarium of St. Michael's Hospital in Toronto, or in each participating institution. Mice or platelets from mice were randomly allocated to experimental and control groups. General conditions of mice were monitored, and the blood flow was evaluated before in vivo experiments. Mice with conditions such as dehydration, or reduced activity were excluded from the study. All animal studies were approved by the Animal Care Committees of St. Michael's Hospital, Toronto, Canada, University of Cincinnati, Cincinnati, USA and The Scripps Research Institute, La Jolla, USA.

**Mouse platelet and plasma preparation.** Mice (6–8 weeks old) were anaesthetized and bled from the retro-orbital plexus using heparin-coated glass capillary tubes[29,30]. Blood was collected into a tube containing sodium citrate or ACD (38 mM citric acid, 75 mM trisodium citrate, 100 mM dextrose). PRP was obtained by centrifugation at $300 \times g$ for 7 min. Gel-filtered platelets were then isolated from the PRP using a Sepharose 2B column in PIPES buffer (PIPES 5 mM, NaCl 1.37 mM, KCl 4 mM, glucose 0.1%, pH 7.0). Platelet-poor plasma (PPP) was prepared by centrifugation at $1500 \times g$ for 20 min. The PPP was further centrifuged at $10,000 \times g$ for 5 min to remove the remaining cells.

**Human blood preparation.** Human blood samples were drawn from antecubital veins of healthy volunteers after providing informed consent. Blood was obtained by venipuncture into Li-Heparin Vacutainers and was allowed to rest at 37 °C for 10 min prior to preparation of PRP. Whole anticoagulated blood was spun at $300 \times g$ for 7 min. The PRP was transferred to a fresh tube and stored at 37 °C before experiments. Gel-filtered human platelets were prepared as described above for mouse platelets. Human blood samples with significant hemolysis or visible clot formation were excluded from this study. Human platelets were randomly allocated to recombinant apoA-IV treatment or control groups. All experimental procedures using human blood samples were approved by the Research Ethics Board of St. Michael's Hospital, Toronto, Canada, the Institutional Review Board of the Georgia Institute of Technology, and Guangdong Provincial Hospital of Chinese Medicine, Guangzhou, China.

**Isolation of β3 integrin ligands.** Human β3 integrins were coated on the latex beads and the presence of active β3 integrins on the beads was confirmed by flow cytometry with PAC-1 or anti-β3 integrin anti-sera using a FACScan™ flow cytometer (BD Biosciences, USA). Ligands were precipitated from fractionated, heparin anticoagulated human blood spin filters (Centricon YM-100 and YM-50, Millipore, Canada) according to the manufacturer's protocol. Beads were then washed, resuspended in protein loading buffer, and subjected to 2D electrophoresis. The protein spots of interest were identified by MALDI Quadrupole time-of-flight mass spectrometry.

**Detection of apoA-IV on activated platelets.** PRP was obtained from WT or β3$^{-/-}$ mice and were pre-incubated with anti-β3 integrin mAb M1 (where indicated) for flow cytometry. Resting or activated (20 μM or 50 μM ADP or 5 μg/mL collagen) platelets were fixed with 4% paraformaldehyde at 4 °C. Samples were incubated with goat anti-human apoA-IV antibody (N-20, Santa Cruz), which also detects mouse apoA-IV, and then incubated with fluorescein isothiocyanate

(FITC)-conjugated donkey anti-goat IgG (Santa Cruz) at room temperature in the dark. Phosphate-buffered saline (PBS, 0.2 mL) was added to the sample immediately before acquisition. All samples were analyzed by flow cytometry using a calibrated FACScan™ flow cytometer (BD Biosciences). For Western blot analysis, resting or activated (0.5 U/mL thrombin) gel-filtered mouse platelets ($4 \times 10^7$ to $1 \times 10^8$) were lysed, separated by SDS-polyacrylamide gel electrophoresis (PAGE), transferred onto polyvinylidene fluoride (PVDF) membrane, and probed with polyclonal goat anti-human apoA-IV antibody and rabbit anti-goat IgG HRP conjugated antibody (Bio-Techne). Following apoA-IV detection, actin was probed as a loading control after stripping the blots and the actin bands were visualized using HRP conjugated goat anti-mouse actin antibody (Supplementary Fig. 16).

**Protein expression and purification.** Recombinant human and mouse apoA-IV were expressed in *Escherichia coli* (*E. coli*) and purified by immobilized metal affinity chromatography (IMAC) columns (His GraviTrap, GE Healthcare)[67]. Briefly, full-length human and mouse apoA-IV were subcloned into the pET30 *E. coli* expression plasmid containing 6× His tag and tobacco etch virus (TEV) protease cleavage site. The uncut protein was expressed by 1-thio-β-D-galacto-pyranoside induction and purified by affinity chromatography utilizing the vector's His tag and Ni$^{2+}$ resin columns followed by overnight dialysis in PBS buffer at 4 °C. The tag was then cleaved using the TEV protease overnight at 4 °C and the cleaved tag was separated using the Ni$^{2+}$ columns and collecting the eluent containing pure cut protein.

**Site-specific biotinylation of apoA-IV.** Site-specific cysteine incorporation for biotinylation was accomplished by site-directed mutagenesis to introduce a cysteine residue at the C-terminus for plasmid construction[68]. The target gene fragment was first PCR amplified and subcloned into pET 30 *E. coli* expression vector. Using human apoA-IV as the template, the target gene fragment was amplified by PCR with upstream primers containing TEV protease site (GAGAACCTGTACTTCCAGGGCGAG) with an Nco I site, and a downstream primer including C-terminal linker GlyGlyCys (GGGGGCTGC) with a hind III site. The final construct (TEV-apoAIV-GGC) was then inserted into a pET 30 expression vector using restriction endonucleases Nco I and Hind III. All mutations were verified through sequence analysis. The expression plasmid TEV-apoA-IV-GGC was next transformed into *E. coli* BL21 cells. Protein expression and purification were performed as described above. For in vitro biotin labeling, the C-terminal biotinylation of apoA-IV-GGC was performed by adding maleimide-PEG2-Biotin (Thermo Scientific) into apoA-IV-GGC in PBS (pH7.4) for 30 min at room temperature to produce a C-terminal biotinylated apoA-IV. SDS-PAGE and Western blot analysis were used to verify apoA-IV biotinylation.

**Biomembrane-Force-Probe detection of apoA-IV-αIIbβ3 integrin interactions.** Purified proteins (i.e., Fg and αIIbβ3) were covalently pre-coupled with maleimide-PEG3500-NHS (JenKem, USA) in carbonate/bicarbonate buffer (pH 8.5). To coat proteins onto the glass beads, 2 μm silanized borosilicate beads (Thermo Scientific) were first covalently coupled with mercapto-propyl-trimethoxysilane (Sigma-Aldrich), then covalently linked to maleimide modified streptavidin (SA, Sigma-Aldrich) and proteins of interest in PBS (pH 6.8) by overnight incubation[33,34]. To coat beads with apoA-IV, SA alone pre-coupled beads were prepared as above. Subsaturating biotinylated apoA-IV was coupled to SA beads by 2 h incubation at room temperature. After resuspending in PBS with 1% BSA, beads were ready for immediate use in BFP experiments. Please refer to the published protocols for details[33,34].

For the BFP adhesion frequency assay, in each BFP cycle, the target (a receptor-bearing bead, cell, or platelet, i.e., αIIbβ3) was driven to approach and contact the probe (a ligand-bearing bead, e.g., apoA-IV or Fg) with a ~15 pN compressive force for a certain contact time ($t_c$) (0.2 s by default, 0.1–1 s for adhesion frequency assay, 1 s for competition assay) that allowed for bond formation and then retracted for adhesion detection. During retraction, tensile force signified an adhesion event and no tensile force indicated a non-adhesion event. Adhesion and non-adhesion events were enumerated to calculate an adhesion frequency ($P_a$) in 50 cycles for each probe-target pair. As $P_a$ depends on $t_c$, the 2D effective on-rate ($A_c k_{on}$) and 2D off-rate ($k_{off}$) can be derived by fitting the $P_a$ vs $t_c$ curve with the model $P_a = 1 - \exp\{-m_r m_l A_c k_{on}[1 - \exp(-k_{off} t_c)]/k_{off}\}$ [34], where $m_r$ and $m_l$ are the respective surface densities of receptors and ligands measured by flow cytometry. The 2D effective affinity ($A_c K_a$) is calculated as $A_c k_{on}/k_{off}$. Three to five probe-target pairs were measured at each $t_c$. For the competition assay, $P_a$ of Fg-αIIbβ3 interaction was measured with different concentrations of apoA-IV in solution.

**Flow cytometry detection of platelet-bound Fg on human platelets.** Resting human gel-filtered platelets ($1 \times 10^8$) were incubated with increasing doses of human recombinant apoA-IV or BSA control, then incubated with FITC-conjugated human Fg (15 min, room temperature). Platelet activation was induced by TRAP (250 μM, 5 min, room temperature) in the dark. The samples were then fixed using 4% PFA. Fg binding to the platelet surface was analyzed by flow cytometry.

**Flow cytometry detection of P-selectin expression upon platelet activation**. Resting WT or β3 integrin$^{-/-}$ mouse gel-filtered platelets were incubated with recombinant mouse apoA-IV (160 μg/mL) or the same molar concentration of albumin for 10 min. To minimize αIIbβ3 integrin outside-in signaling induced by ligand-αIIbβ3 interaction, particularly following platelet granule release (e.g., Fg, VWF, fibronectin, and multimerin) that causes platelet aggregation, low platelet concentrations ($4 \times 10^5$/mL) were used. After incubation, the platelets were treated with thrombin (2 U/mL), collagen (2.5 μg/mL), and ADP (40 μM) for 2 min. P-selectin expression was then detected with flow cytometry.

**In vitro platelet aggregation assays**. PRP and PPP from human and mice (apoA-IV$^{-/-}$, apoA-IV$^{+/-}$, apoA-IV$^{+/+}$ control, apoA-IV transgenic mice and WT mice) were obtained by centrifugation of sodium citrate anticoagulated whole blood. Platelet concentration in PRP or PIPES buffer was adjusted to $3 \times 10^8$ platelets/mL with the use of autologous PPP or PIPES buffer[27,30,41]. Platelet aggregation was performed at 37 °C with a sample stir speed of 1000 rpm using a computerized aggregometer (Chrono-Log Corp, Havertown, USA). Platelet aggregation in mouse PRP was stimulated by 1 μM ADP, 1 μg/mL collagen, and 250 μM TRAP. Platelet aggregation in human PRP was stimulated by 2.5 μM ADP and 5 μg/mL collagen. Platelet aggregation in gel-filtered platelets was stimulated by 10 μg/mL collagen and 0. 5U thrombin. For platelet aggregation after switching plasma from the opposite genotype[40], gel-filtered apoA-IV$^{-/-}$ and apoA-IV$^{+/+}$ control platelets were suspended in PPP ($3 \times 10^8$ platelets/mL) from either the same or opposite genotype, and aggregation was induced by 250 μM TRAP. In another set of experiments, gel-filtered platelet aggregation in PIPES buffer was induced by 250 μM TRAP. Platelet aggregation induced by various agonists was also measured after adding indicated doses of human or mouse recombinant apoA-IV (incubation for 2 min). The change in light transmission was monitored and recorded for at least 10 min.

**Depletion of human plasma apoA-IV**. Goat anti-human apoA-IV antibody (C-20, Santa Cruz) was used to deplete apoA-IV in human PPP. Antibody was first coated on the protein G Sepharose beads. The same amount of non-specific goat IgG (Santa Cruz) was used as control. Human PPP was prepared as described above, and added to the beads. The eluate was collected and used for western blot (Supplementary Fig. 17) and platelet aggregometry. For platelet aggregometry, one third volume of normal human PRP was combined with two third volumes of apoA-IV-depleted human PPP or non-specific goat IgG-treated control PPP.

**Site-directed mutagenesis**. Depletion and point mutations of human and mouse apoA-IV were created by site-directed mutagenesis[37]. Human or mouse apoA-IV DNA was ligated into the pET30 expression vector using the NcoI and Hind III sites with a cleavable N-terminal 6×His tag as described above. We generated the single mutants Δ1–38 apoA-IV, Δ336–376 apoA-IV, D5E apoA-IV, D13E apoA-IV and double mutants 39–335 (Δ1–38, 336–376 apoA-IV), DM (D5E, D13E apoA-IV). Point mutagenesis was performed in the expression vector using the Quick Change Site-Directed Mutagenesis Kit (Agilent Technologies, Canada). All mutations were verified through sequence analysis.

**Circular dichroism spectroscopy (CD) and thermal denaturation assays**. WT and DM apoA-IV recombinant proteins were purified by size exclusion chromatography on a BioRad SEC650 column equilibrated in PBS. Protein concentration and purity were quantified by A280 and SDS-PAGE analysis, respectively. CD spectral scans and thermal melts of WT and DM apoA-IV (concentrations ranging from 0.4 to 0.5 mg/mL) were acquired on a Jasco J-810 spectropolarimeter using a 1 mm quartz cuvette (Helma). CD wavelength scans were collected between 190 and 250 nm and averaged over five accumulations. Raw data was then converted to molar ellipticity (deg.cm$^2$.dmol$^{-1}$). Thermal denaturation assays were carried out at a single wavelength (209 nm) by monitoring the change in ellipticity as the temperature was raised from 20 °C to 90 °C at a rate of 5 °C/min. All thermal denaturation data were baseline corrected, normalized between 0 (folded) and 1 (unfolded) and were fit to a nonlinear biphasic sigmoidal curve in GraphPad (GraphPad Software, San Diego, CA, USA).

**Ex vivo perfusion chamber assays**. Rectangular microcapillary chambers (ibidi channel slides, ibidi GmbH) were coated with 100 μg/mL type-I collagen (Nycomed) or 50 μg/mL human Fg (Sigma-Aldrich) overnight at 4 °C[30,31,45]. For murine studies, whole blood was collected from anaesthetized apoA-IV$^{-/-}$, apoA-IV$^{+/+}$ control, or WT control mice (C57BL/6 J) with heparin (15 U/mL). Blood was then fluorescently labeled with DiOC6 (1 μM, Sigma; 10 min, 37 °C). ApoA-IV$^{-/-}$ or apoA-IV$^{+/+}$ control mouse blood was perfused over the collagen-coated surface using a syringe pump (3 min; 300 s$^{-1}$ and 1800 s$^{-1}$; Harvard Apparatus, USA). WT control blood (treated with same volume of PBS), recombinant apoA-IV-treated blood (160 μg/mL), and DM apoA-IV-treated blood (160 μg/mL) were perfused at shear rates from 300 s$^{-1}$ to 5000 s$^{-1}$. We performed human blood perfusion chamber assays as described above. Platelet aggregation and thrombus formation or platelet adhesion to Fg were recorded in real-time under a Zeiss Axiovert 135-inverted fluorescence microscope (32 X-W objectives).

Quantitative dynamics of platelet fluorescence intensity were acquired by Slide-Book software (Intelligent Imaging Innovations Inc., USA).

**Intravital microscopy thrombosis models**. The experiments on intravital microscopy thrombosis models were performed after injuries induced by ferric chloride (FeCl₃)[27,29,41,45] or laser[30,31,41,43]. The person conducting the experiment was unaware as to the genotype or treatments received by the mice.

For the mesenteric arterial thrombosis model, thrombus formation in mesenteric arterioles was monitored in 3- to 4-week-old mice[27,29,41,45]. ApoA-IV$^{-/-}$, apoA-IV$^{+/+}$, WT control, and WT mice treated with recombinant or DM apoA-IV were used. Mice were injected with donor-matched fluorescently labeled platelets and visualized under a Zeiss Axiovert 135-invertedfluorescent microscope (Zeiss, Germany). Briefly, blood was collected into ACD from the donor-matched mice. Gel-filtered platelets were prepared and labeled with Calcein AM (1 mg/mL, Invitrogen, Canada) at room temperature for 20 min. Platelets were then injected into the experimental mice via the tail vein with control saline buffer, mouse recombinant apoA-IV (9 μg/g) or DM apoA-IV (9 μg/g). Mice were then anesthetized and the mesentery was externalized. A single mesenteric arteriole of 100–120 μm diameter was chosen and injury was induced by topical application of 30 μL of 250 mM ferric chloride. The time required for formation of the first 20 μm thrombus and the time to complete vessel occlusion was recorded. Images of thrombus formation and dissolution were visualized with a fluorescence microscope.

For the laser-induced cremaster arterial thrombosis model[30,31,41,43], apoA-IV$^{-/-}$, apoA-IV$^{+/+}$ control, and WT control mice (male, 6–8 weeks old) were anesthetized and a tracheal tube was inserted to facilitate breathing. The cremaster muscle was prepared under a dissecting microscope and superfused throughout the experiment with preheated bicarbonate-buffered saline. Antibody, control saline buffer, mouse recombinant apoA-IV (9 μg/g) or DM apoA-IV (9 μg/g) were administered where indicated by a jugular vein cannula. Platelets were labeled by the rat anti-mouse CD41 antibody (Leo.A1; EMFRET Analytics, Germany; 0.1 μg/g) injection. Multiple independent upstream injuries were induced on a cremaster arteriole using an Olympus BX51WI microscope with a pulsed nitrogen dye laser. The dynamic accumulation of fluorescently labeled platelets within the growing thrombus was captured and analyzed using Slidebook software.

**Ferric chloride-induced carotid artery thrombosis**. Experiments were performed in >8-week old, 20–35 g, isoflurane-anesthetized C57BL/6J mice of both genders[45,46]. Ten minutes before inducing arterial injury, WT or mutant apoA-IV (9 μg/g), or an equal volume (100 μL) of PBS were injected through a catheter into the right jugular vein. After measuring the baseline blood flow rate (volume/time) for 20 s (s), the left common carotid artery was injured by applying fresh FeCl₃•6H₂O solution (0.7 μL, 8%, 0.30 M Fe$^{3+}$ ion) onto the exposed adventitia for 180 s. The vessel was then rinsed with warm physiologic saline solution and the flow rate was measured for an additional 1800 s (1980 s total from beginning of injury). Three parameters were calculated: (1) time to first occlusion (O-1), until the flow rate was ≤0.1 mL/min (maximum zero flow error of the probe); (2) time to stable occlusion (O-S), until the flow rate was ≤0.1 mL/min lasting at that level for ≥10 min; and (3) flow index (FI), the ratio of the volume of blood flowed through the artery after-injury (calculated by integrating post-injury flow rate values sampled every second for 1800 s) to the volume flowed through in an uninjured vessel (calculated from the average pre-injury flow rate). For the carotid artery thrombosis model in apoA-IV$^{-/-}$ and apoA-IV$^{+/+}$ control mice, the right carotid artery was dissected following anesthetization and held with a miniature Doppler flow probe (TS420 transit-time perivascular flowmeter, Transonic Systems Inc., USA). Carotid artery injury was induced with a strip of Whatman filter paper saturated with 10% ferric chloride. Blood flow was monitored until complete vessel occlusion was observed.

**Bleeding time assay**. ApoA-IV$^{-/-}$, apoA-IV$^{+/+}$, and WT mice (6–8 weeks old) were injected with recombinant apoA-IV where indicated via the tail vein 40 min before injury. Mice were anesthetized and maintained at 37 °C on a heating pad during the experiment. The tip of the tail (5 mm) was cut off with a sharp scalpel, and the tail was immediately placed into warm saline at 37 °C. Bleeding time was recorded as the time to cessation of blood flow (bleeding stopped for >10 s). Blood loss was calculated by counting the red blood cells in the PBS fraction.

**Postprandial platelet function assay**. C57BL/6 J WT, apoA-IV transgenic (apoA-IV-Tg), apoA-IV$^{-/-}$, apoA-IV heterozygous (apoA-IV$^{+/-}$), and apoA-IV$^{+/+}$ mice (6–8 weeks old) were fasted for 12 h, followed by a HFD for 3 h. Blood was then collected from these mice following fasting or HFD condition. Gel- filtered platelets ($2.5 \times 10^8$/mL) and PPP were prepared respectively. To minimize interference in the light transmission of PRP, which is commonly caused by postprandial chylomicrons and other lipid particles[69], 125 μL gel-filtered platelets were incubated with its PPP (125 μL) for 20 min under 37 °C. The same volume of PPP and PIPES buffer were used as blank control. Platelet aggregations were then induced by collagen (10 μg/mL).

**Enzyme-linked immunosorbent assay to detect αIIbβ3 binding**. A 96-well plate (Nunc MaxiSorp) was coated with purified human αIIbβ3 or control proteins (BSA and β3$^{-/-}$ platelet lysate) by incubation of 0.01 μg/μL protein in binding buffer

(TRIS buffered saline with 0.05 % TWEEN-20 and 1 mM each of $MgCl_2$, $MnCl_2$ and $CaCl_2$) at 4 °C overnight. Incubation of 3% skim milk (ED Millipore) and 2% TWEEN-20 for 1 h at 37 °C was used for blocking. For Fg binding experiments, wells were incubated with the appropriate Fg concentration in binding buffer for 1 h at 37 °C. Wells were then incubated with anti-fibrinogen mouse IgG (Sigma Aldrich) followed by incubation with anti-mouse IgG-HRP (Santa Cruz), both at 37 °C for 1 h. For apoA-IV binding experiments the desired protein concentration in binding buffer was added to each well and incubated 1 h at 37 °C followed by the incubation of streptavidin-horseradish peroxidase (HRP) conjugated protein (Abcam), at 37 °C for 1 h. Peroxidase substrate, o-Phenylenediamine dihydrochloride (OPD, Sigma Aldrich) was prepared at 0.4 mg/mL in 0.05 M phosphate citrate buffer at pH = 5.0 with 0.4 μL/mL of 30 % $H_2O_2$. The OPD peroxidase reaction was stopped after 1 h with 2 M $H_2SO_4$ and the absorbance was read at 492 nm. Between incubation steps the plate was washed with copious amounts of TRIS buffered saline with 0.5% TWEEN-20.

IC50 Determination: For the IC50 of apoA-IV; 0, 0.25, 0.5, 0.75, 1, and 5 μM Fg was bound to immobilized αIIbβ3 in the presence of 1 μM apoA-IV in binding buffer. While for the IC50 of Fg; 0, 0.3, 0.7, 1, 2, 5, and 10 μM apoA-IV were each mixed with 1 μM Fg in binding buffer and bound to immobilized αIIbβ3. ELISA absorbance was plotted against inhibitor concentration and the curves were fit to the half maximal inhibitory model, eq. 1[70].

$$y = \frac{E_{\max}}{1 + \frac{IC_{50}}{[I]}}.$$

**Human study to evaluate the correlation between apoA-IV and platelet aggregation**. Twenty healthy volunteers taking no medications (mean (range), 25.8 (20–31) years; body mass index, 23.6 (19.9–29.6) kg/m2; 10 women and 10 men) were recruited in this study. All participants gave written informed consent in accordance with the Declaration of Helsinki.

The meals, sleep, physical activity, room temperature, and light the volunteers received were all controlled. Blood samples were collected prior to meals via venipuncture at 12:00, 18:00, 24:00, and 06:00 across a 24-h period. PRP and PPP were prepared as described above. Plasma samples to be assayed for the concentration of apoA-IV were stored at −80 °C until analyzed by ELISA. Platelet aggregation was induced by ADP (6 μM) and assays were performed immediately using fresh PRP and PPP within 1 h after blood samples were collected.

**Enzyme-linked immunosorbent assay to measure apoA-IV plasma level**. Plasma apoA-IV concentrations were determined using an ELISA that employs affinity purified goat anti-human apoA-IV polyclonal antibody as the capture antibody (apoA-IV N-20; Santa Cruz). ApoA-IV N-20 was raised against a peptide mapping near the N-terminus of apoA-IV of human origin. Recombinant human apoA-IV proteins with known concentrations (determined by Nanodrop) served as a standard. ApoA-IV N-20 (1 μg/mL) was coated on the surface of ELISA plates (Nunc MaxiSorp) at 4 °C for overnight. 3% Skim Milk (Millipore) with 2% Tween-20 in PBS buffer were used for blocking the wells at 37 °C for 1 h. We then incubated recombinant human apoA-IV or human plasma samples (100 times dilution) at 37 °C for 3 h, followed by incubation with the detection antibody, apoA-IV G-8, a mouse monoclonal antibody raised against a peptide mapping near the C-terminus of apoA-IV of human origin (2ug/mL; Santa Cruz) at 37 °C for 2 h. The mouse IgG kappa binding protein (m-IgGκ BP) conjugated to HRP (1ug/mL; Santa Cruz) was then incubated in the wells at 37 °C for 1 h. OPD substrate and the stop solution were used as described above. Three to five times washing were performed between each step. Read absorbance at 492 nm and 650 nm in a plate reader.

**Statistical analysis**. Data are presented as mean ± SEM. Statistical significance was assessed by unpaired, two-tailed Student's t-test and non-parametric Kruskal–Wallis one-way analysis of variance followed by Dunn's test for multiple paired comparisons. The sample size of each experiment was estimated based on our previous experience and the publications from others[32,33,37,47,50].

## Data availability

The data that support the findings of this study are available from the first author and the corresponding author upon reasonable request.

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

## Acknowledgements

The authors thank Dr. Richard O. Hynes for providing the β3 integrin gene-deficient mice, Dr. Karen Reue for providing the apoA-IV transgenic mice, and Dr. Wilbur Lam and his lab for blood collection in BFP experiments. The authors thank Dr. Alan Nurden for valuable suggestions and comments during the manuscript preparation. The authors also thank Dr. William P. Sheffield, June Li, Xun Fu, and Alexandra Marshall for editing the manuscript. This work was supported in part by a grant-in-aid from the Heart and Stroke Foundation of Canada (Ontario); Canadian Institutes of Health Research (MOP 119540, and MOP 97918); Equipment Funds from St. Michael's Hospital, Canadian Blood Services, and the Canada Foundation for Innovation. Dr. Xiaohong Ruby Xu is a recipient of the Heart and Stroke/Richard Lewar Centre of Excellence Studentship award, Meredith & Malcolm Silver Scholarship in Cardiovascular Studies, and China National Scholarship award. Dr. Yiming Wang is a recipient of the Ph.D. Graduate Fellowship from Canadian Blood Services and Meredith & Malcolm Silver Scholarship in Cardiovascular Studies. Dr. Lining Ju is a recipient of National Heart Foundation of Australia postdoctoral fellowship (101285), CSANZ-BAYER Young Investigator Research Grants, the Royal College of Pathologists of Australasia Kanematsu research award, Diabetes Australia research award (G179720) and Sydney Medical School early-career researcher kickstart grant. Dr. Joseph Wuxun Jin was a recipient of the Canadian Blood Services postdoctoral fellowship award and Heart and Stroke Foundation of Canada research fellowship award. Dr. Hong Yang was a recipient of the Heart and Stroke/Richard Lewar Centre of Excellence Studentship award and a Canadian Blood Services postdoctoral fellowship award. Dr. Yan Yang was a recipient of the Canadian Blood Services postdoctoral fellowship award. Work performed by L.J., Y.C. and C.Z. (HL-132019) and by P.M. and Z.M.R. (HL-031950) was supported by the National Institutes of Health of the USA, Bethesda, MD.

## Author contributions

X.X. constructed plasmids, generated recombinant proteins, designed and performed platelet aggregation, perfusion chamber, high fat diet platelet functional experiment, intravital microscopy studies, circular dichroism spectroscopy, thermal denaturation assays, and clinical studies with the assistance from P.C., L.X. P.K., W.F., and J.E.L., analyzed data and wrote the draft manuscript. Y.W. and R.A. designed, performed, and analyzed the intravital microscopy studies and flow cytometry P-selectin studies. L.J. performed and analyzed Biomembrane-Force-Probe studies with assistance from Y.C., S. P.J., and C.Z. C.M.S. designed, performed, and analyzed the flow cytometry and bleeding time assays. J.W.J. performed and analyzed the platelet aggregation, intravital microscopy studies. H.Y. performed studies in isolating β3 integrin ligands, 2D electrophoresis, and identified apoA-IV with Y.S. and T.C. using mass spectrometry. M.A.D.N performed ELISA assays. Y.Y. studied thrombosis in β3 integrin heterozygous mice. R.C.G and M.X studied apoA-IV function in thromboelastogram assay and shear-induced PS exposure assay, respectively. H.Z. constructed plasmid of biotinylated apoA-IV and generated proteins. D.Z. performed platelet aggregation assays and generated recombinant proteins. N.C. performed platelet activation assays. S.Y performed the western blot with the assistance from Y.W. and M.A.D.N. G.Z. generated anti-β3 integrin monoclonal antibody M1. J.S. performed and analyzed platelet cAMP studies. Z.M.R. and P.M. designed, performed, and analyzed the studies to establish the effect of apoA-IV on ferric chloride-induced carotid artery thrombosis in mice. G.L., P.W.C., M.L.R., K.A., and J.F. were involved in study design and data analysis. P.T. performed studies in measuring plasma lipid and cholesterol components. W.S.D. provided the plasmid of apoA-IV and mutant

proteins. H.N. is the principal investigator who designed the research, analyzed the data, and wrote the manuscript. X.X., Y.W., R.A., L.J., C.M.S., and J.W.J contributed equally to this work.

## Additional information

**Competing interests:** The authors declare no competing interests.

