## [Peer Review File · Nature Communications]

ANSWERS TO REVIEWERS' COMMENTS:

Reviewer #2:

Remarks to the Author:

The authors have clearly made a conscientious effort to respond to the critique of their revised paper. I have some comments regarding the authors's responses to my previous concerns.

1. I don't entirely agree with authors' contention regarding differences between hemostasis and thrombosis. Much of this is based on mouse models whose correspondence of the *in vivo* situation in humans is unclear to say the least. Further, I like to know which effective anti-thrombotics don't significantly compromise hemostasis. The major complication of every anti-thrombotic that I know of and use is bleeding.

Response:

Although the major complication of anti-thrombotic agents is bleeding, we do not think this is the evidence to negate the important difference between hemostasis and thrombosis. Clinically, the key to optimal use of any anti-thrombotic agents is to adjust the dosage to reach a level that could efficiently control thrombotic events without causing significant bleeding risks (e.g., uncontrolled bleeding from a minor wound) for each individual patient. Therefore, even in the clinical setting, the therapeutic range, albeit sometimes narrow, does exist to enable treatment of thrombosis without significantly compromising hemostasis.

"Hemostasis" is a physiological process but thrombosis is pathological. Thrombosis is the process that blocks the blood flow through the vasculature, e.g., occludes the vessel lumen, leading to functional impairment or damage of tissue or organs. However, the key of hemostasis, in most cases, is to mend the opening on the vessel wall and block the leakage of blood cells, which do not require the complete occlusion of the vessel lumen. The levels of platelet activation and fibrin deposition in the core of a hemostatic plug at the site of injured vessel wall are far higher and the shear stress is far lower than at the apex of thrombotic plug (Stalker TJ, et al. *Blood*. 2013 Mar 7;121(10):1875-85. and Xu XR, et al. *Thromb J*. 2016 Oct 4;14(Suppl 1):29.), making a thrombotic process more susceptible to inhibitory agents.

Therefore, although these two biological processes share many common pathways, many studies (not only just mouse models), such as the report in baboons (Coller BS, et al. *Circulation*. 1989;80:1766-74), clearly demonstrated that thrombosis is more sensitive than platelet aggregation, and the platelet aggregation is more sensitive than bleeding to antagonists of α IIB β 3 (*i.e.* thrombosis > platelet aggregation > hemostasis), which is also the basis for most anti-thrombotic therapies that efficiently decrease thrombosis without significantly compromising hemostasis.

Our data are consistent with these studies, we did find that apoA-IV, at an extremely high dose (>5 times of physiological level), prolonged the tail bleeding time. However, at physiological plasma concentration, apoA-IV is able to reduce thrombosis without prolonging bleeding time, which is similar to the clinical treatment of thrombosis with an anti-thrombotic agent at the optimal dose or at the lower doses to prevent thrombosis. ApoA-IV is therefore an important preventive and protective factor against thrombosis and cardiovascular diseases.

2. The major effect of increasing ApoA-IV on platelet function, best seen when ADP is the platelet agonist, is impaired second wave (i.e., secondary) aggregation. Further, the ability of increasing collagen concentrations to “overcome” the effect of exogenous ApoA-IV (Supplement Figure 4b) suggests that ApoA-IV affects platelet sensitivity to stimulation.

Response:

We agree that apoA-IV predominantly decreases the later phase (i.e., second wave) of platelet aggregation (Fig. 3-4). However, apoA-IV slowed the progression of platelet aggregation in most of our experiments (Fig. 3a-e, Fig. 4e-h), although the initial rate of aggregate formation is only moderately affected. This is particularly not easy to be seen when the X axes are compressed to incorporate the full 15-minute platelet aggregation curve into the figures.

We showed that, apoA-IV has lower affinity for α IIb β 3 than fibrinogen (Fig. 2a and c) and does not directly alter platelet activation (Supplementary Fig. 6a-c), and thus can only mildly slow down the initial rate of platelet aggregation. However, the inhibition of fibrinogen or other multivalent ligands binding to α IIb β 3 integrin by apoA-IV can further attenuate α IIb β 3 outside-in signaling, which can subsequently decrease platelet granule release (Supplementary Fig. 6b) and the 2nd wave of platelet aggregation. Thus, the inhibitory effect of apoA-IV appears to be more obvious in the later phase of platelet aggregation through the inhibition of α IIb β 3 integrin. However, as we discussed in the paper, we cannot completely exclude whether apoA-IV can affect platelet sensitivity to stimulation via other pathway(s).

It is understandable that, for most platelet inhibitors, the inhibition efficacy decreases as the level of platelet agonist increase. The likely reasons include platelet shape change and pro-thrombotic ligand release from platelet α granules, particularly those high affinity multivalent ligand complex released from α granules (Reheman A, et al. J Thromb Haemost. 2005 May;3(5):875-83.; Reheman A, et al. Thromb Res. 2010 May;125(5):e177-83.; and Reheman A, et al. Blood. 2009 Feb 19;113(8):1809-17.).

These high avidity of ligands can reduce the accessibility of inhibitors to α IIb β 3 integrin. This may explain the *slight trend of the decreased* inhibitory effect of apoA-IV when the concentration of collagen was increased from 1.25 μ g/mL to 2.5 μ g/mL. However, further increasing the concentration of collagen cannot “overcome” the inhibitory effect apoA-IV (Supplementary Fig. 5b), suggesting that

apoA-IV is a competitive ligand and plays an important inhibitory role in α IIB β 3 integrin-mediated platelet aggregation.

3. On Page 6, the authors state that the 2D affinity of ApoA-IV for α IIB β 3 is 40% that of fibrinogen. What is the relationship of 2D affinity to the affinity of soluble fibrinogen and ApoA-IV determined kinetically?

Response:

The 2D affinity and 3D affinity measurements represent ApoA-IV binding in different scenarios. The former describes when the ligand is immobilized on the surface whereas the later reports when the ligand is in solution. As we previously demonstrated in a T-cell receptor system (Huang J, et al. Nature. 2010 Apr 8; 464(7290):932-6.), they are closely associated concepts but may not directly correlate to each other. The 2D affinity affects platelet adhesion to the vessel wall or immobilized surface (Supplementary Figs. 2a-b), and the 3D affinity determines its inhibitory effect on platelet aggregation in solution or in blood as we demonstrated in vitro and in vivo.

4. Again, was α IIB β 3 on CHO cells activated? “Low affinity” ligand binding to α IIB β 3 is really not specific binding.

Response:

The α IIB β 3 integrins on CHO cells are mainly inactive, but there are approximately 11% of CHO cells are positive for active α IIB β 3 specific antibody binding (Mekrache M, et al. Br J Haematol. 2002 Dec;119 (4): 1024–1032), suggesting small portion of α IIB β 3 molecules are in active conformation. Furthermore, different chemical (e.g. divalent cations) (Mekrache M, et al. Br J Haematol. 2002 Dec; 119(4): 1024–32.; Gailit J, et al. J Biol Chem. 1988 Sep 15;263(26):12927-32.; and Ni H, et al. J Biol Chem. 1998 Apr; 273(14): 7981-7987.) and physical (e.g. ligand or antibody bindings) (O'Toole TE, et al. Cell Regul. 1990 Nov; 1(12): 883-893. and Lin FY, et al. J Biol Chem. 2016 Feb 26;291(9):4537-46.) stimulations can switch the inactive α IIB β 3 from low to high affinity states.

Biomembrane Force Probe (BFP) is a technique that can sensitively detect the receptor-ligand interactions, and was therefore employed to detect the apoA-IV binding to integrins on CHO cells and resting platelets. Following platelet activation and conversion of α IIB β 3 to the high affinity state, the adhesion frequency between apoA-IV and the platelet was markedly increased (Fig. 1c). Importantly, apoA-IV bindings to α IIB β 3 integrin in these experiments were completely inhibited by a blocking anti- β 3 integrin monoclonal antibody M1 (Reheman A, et al. Blood. 2009 Feb 19;113(8):1809-17.) (Fig. 1c-e). These data demonstrate the specificity of the direct interaction between apoA-IV and α IIB β 3 integrin.

5. The observation that two Asp residues (D5 and D13) required for ApoA-IV binding

is interesting but I don't think it is entirely accurate to equate this in some way to RGD binding. It is my understanding that the separation of the + charged Arg and – charged Asp of RGD is important for motif binding and for selectivity among various RGD-binding integrins.

Response:

While the interaction between the non-RGD dependent fibrinogen (there is no + charged Arg and – charged Asp in the HHLGGAKQAGDV peptide of fibrinogen γ chain C terminus) and α IIb β 3 integrin is well characterized, the binding mechanism of other non-RGD containing integrin ligands is not fully understood. Future study (e.g., co-crystallization of apoA-IV and β 3 integrin) may be able to further elucidate the underlying mechanism.

6. The authors are to be congratulated for embarking on the human studies. Clinical studies are difficult enough, but doing them around the clock adds to the difficulty. However, while the reciprocal changes in platelet aggregation and ApoA-IV concentrations support the author's hypothesis, the magnitude of the effects are a concern. Thus, the increase in platelet aggregation between the hours of 6:00 and 24:00 was 0.1 fold, although significant statistically, was essentially trivial and there was only 0.2 fold decrease in the concentration of ApoA-IV over the same time period.

Response:

We appreciate the comments from the reviewer and the clinical studies and doing the experiments around clock are indeed very difficult. We are, however, very pleased that platelet aggregation and ApoA-IV concentrations is consistent with our hypothesis and the observed cardiovascular events. We agree with the reviewer that the fold change of apoA-IV concentration is not very high, which may be partially due to the ELISA assay since it has not been fully optimized and no commercial kit is currently available. With further optimization to reduce the background, the fold changes measured during the circadian rhythm would be more obvious. Notably, as also discussed in the paper, partial inhibition of α IIb β 3 integrin can decrease α IIb β 3-outside in signaling and attenuate platelet further activation, which may significantly affect the second wave (late stage) platelet aggregation. In addition, we cannot exclude other factors that may independently or synergistically interact with apoA-IV to regulate platelet aggregation during the circadian rhythm.

7. Although the data are convincing that ApoA-IV can bind to α IIb β 3 and that it can compete with fibrinogen for binding, the notion that this has physiologic and pathophysiologic relevance I think remains tenuous for the following reasons. Neither physiologic α IIb β 3 ligands such as fibrinogen nor ApoA-IV bind to α IIb β 3 on circulating platelets and only bind after platelets are stimulated. The ApoA-IV concentration in plasma is at least 1-2 orders of magnitude less than fibrinogen and the authors state the affinity of ApoA-IV for α IIb β 3 based on their 2D measurements

is 40% that of fibrinogen. Thus, it is hard to conceive that ApoA-IV could out-compete fibrinogen in vivo. Further, in the Discussion, the authors state that endogenous ApoA-IV blocks 20-40% of the interaction between fibrinogen and α IIB β 3. But experience with exogenous high affinity α IIB β 3 antagonists indicates that 80% α IIB β 3 blockade is needed to meaningfully inhibit platelet aggregation. Finally, as I said last time, the platelets of Glanzmann thrombasthenia heterozygotes expressing 50% of the normal amount function normally.

Response:

We agree that around 80% of α IIB β 3 blockade is needed to “**completely**” inhibit “hyperactive” platelet aggregation of patient with myocardial infarction (Peter K, et al. Circulation. 2000 Sep 26; 102(13):1490-6). However, completely abolishing platelet aggregation is not the goal of anti-thrombotic medications due to a high bleeding risk. Consistently, we are not proposing to “completely” block platelet aggregation with apoA-IV, but to decrease platelet aggregation through blockage of a proportion of α IIB β 3 integrin on the platelet surface. As shown in the same report (Peter K, et al. Circulation. 2000 Sep 26; 102(13) 1490-6): 1490-6), when around 50% of platelets are bound by abciximab, there is already a significant decrease of platelet activity. In another report from Dr. Barry Coller’s group, 22% of α IIB β 3 occupancy is able to markedly inhibit thrombosis in a primate model (Coller BS, et al. Circulation. 1989 Dec;80(6):1766-74.).

It is notable that although fibrinogen concentration (~1.5-4 mg/mL) is approximately 10 times higher than apoA-IV (~0.15-0.37mg/mL), the molecular weight of fibrinogen (~340kDa) is also approximately 7.4 times higher than apoA-IV (~46kDa), therefore the molar ratio is close. Furthermore, considering the two fibrinogen binding sites are simultaneously required for two α IIB β 3 molecules on two platelets during platelet aggregation, but only one apoA-IV- α IIB β 3 occupancy is sufficient to stop the bridge between the adjacent platelets, the inhibitory effect of apoA-IV should not be underestimated.

Furthermore, “Neither physiologic α IIB β 3 ligands such as fibrinogen nor ApoA-IV bind to α IIB β 3 on circulating platelets and only bind after platelets are stimulated” may not be completely correct since significant interactions between fibrinogen and α IIB β 3 integrin occur on the circulating platelets in vivo, which is required for fibrinogen internalization (Ni H, et al. Blood. 2003 Nov 15;102(10):3609-14. and Yang H, et al. Blood. 2009 Jul 9;114(2):425-36.) as we all can see fibrinogen inside all human as well as animal platelets.

We think, it is reasonable to propose that blockage of 20-40% of the interaction between fibrinogen and α IIB β 3 by apoA-IV could reduce (but not completely abolish) platelet aggregation and it is subsequent outside-in signaling. Again, in our opinion, realizing the difference between pathological thrombosis and physiological hemostasis is crucial to understand the important role of apoA-IV in thrombosis.

Regarding Glanzmann thrombasthenia heterozygote patients, we agree that they do not have overt bleeding symptoms. However, per our communication with Dr. Alan T. Nurden and other experts in the field, no report is available on the thrombotic risk of the Glanzmann thrombasthenia heterozygotes. Our animal studies (Supplemental Fig. 14) also showed that $\beta 3$ integrin heterozygote mice, which express about 50% $\alpha \text{IIb}\beta 3$ on platelets, exhibit no obvious bleeding disorder but significantly decreased platelet thrombus formation in vitro and in vivo. These findings suggest that Glanzmann thrombasthenia heterozygote patients may have reduced thrombotic risks. Future clinical studies are required to analyze the incidence of heart attack and stroke in these patients and compare it to the normal population.

In addition, ApoA-IV possesses several unique features to balance the pro-thrombotic risks as demonstrated/presented in this paper:

1) ApoA-IV is an abundant plasma protein that can be quickly synthesized after food intake. Through antagonizing platelet integrin $\alpha \text{IIb}\beta 3$, apoA-IV significantly decreases postprandial platelet hyperactivity (Fig. 9). Postprandial spikes in glucose and lipids can generate excess free radicals, induce inflammation and enhance platelet-endothelial cell and platelet-leukocyte interactions, which play an important role in atherogenesis and thrombogenesis. The rapid increase in the secretion of apoA-IV after meals is therefore important to prevent these pathological processes.

2) Through occupancy of $\alpha \text{IIb}\beta 3$ integrin, apoA-IV can also inhibit $\alpha \text{IIb}\beta 3$ integrin outside-in signalling that affects platelet P-selectin (Supplementary Fig. 6b-c) and other protein expression, as well as phosphatidylserine exposure (Supplementary Fig. 7a). ApoA-IV can thus decrease cell-based thrombin generation and blood coagulation (Supplementary Fig. 7b).

3) The anti-thrombotic effect of apoA-IV is escalated following the increasing shear stress (Fig. 5c-d and 6, Supplementary Fig. 10c and 11). This action is critical for inhibiting thrombosis at the sites of stenosis or growing thrombi (Fig. 7 and 8), which prevents vessel occlusion and rescues blood supply to the downstream tissues without compromising hemostasis (Supplementary Fig. 13).

Therefore, through long-term, moderate but significant attenuation of platelet hyperactivity, particularly in the postprandial period, apoA-IV may be physiologically critical to slow down the progression of atherosclerosis, as observed in transgenic mice overexpressing human or murine apoA-IV. Through attenuation of blood coagulation and gaining enhanced anti-thrombotic (i.e. anti- $\alpha \text{IIb}\beta 3$) effects at high shear, apoA-IV also prevents vessel occlusion (i.e. heart attack and stroke). Thus, as an endogenous platelet inhibitor, apoA-IV is an important player in both early (i.e. atherosclerosis) and late (i.e. thrombosis) stages of atherothrombosis. Additionally, since platelets have now been considered versatile cells with diverse biological functions, inhibition of platelet function may have a broad impact on inflammation, immune response, and the recently-highlighted tumorigenesis and tumor metastasis, etc.

Reviewer #3:

Remarks to the Author:

I was asked to review this manuscript after it had already gone through at least one round of review. The data appear relatively compelling that apoA-IV is an endogenous inhibitor of ligand binding to the platelet integrin α IIb β 3 and can inhibit platelet-mediated thrombosis in vitro and vivo.

My major question is regarding the specificity of this observation. ApoA-IV is closely structurally related to apoA-I and apoE. Ideally I would like to see one or both of these well-studied proteins included in at least some of the key in vitro assays as negative controls, indicating that the observed effects are specific to apoA-IV and not a generic property of this class of amphipathic alpha helical proteins that bind to lipoproteins.

Response:

We appreciate the suggestions from the reviewer. It has been reported recently that isolated human apoA-I is also able to inhibit platelet aggregation (Branchford BR, et al. Blood. 2014; 124:4159). However, this is likely through a different mechanism from apoA-IV, since most apoA-I is associated with HDL in blood, and exerts its function through HDL. The role of apoE in inhibition of platelet has also been reported, but it has been found to act through L-Arg: Nitric Oxide pathway (Riddell DR, J Biol Chem, 1997 Jan; 272(1):89-95).

On the other hand, the current study could not completely exclude other mechanisms mediated by apoA-IV, such as possible interactions with HDL, LDL or membrane lipid components, which may synergistically contribute to the inhibitory effect on platelet aggregation and thrombus growth. It is also conceivable that interactions with HDL and other plasma components may increase the sensitivity of apoA-IV to shear stress, which facilitate the accessibility of the N-terminus to α IIb β 3 integrin. These possibilities merit future investigations.

REVIEWER'S COMMENTS

Reviewer #2 (Remarks to Author)

Again the authors have conscientiously responded to my review of the last version of their paper. I have no additional comments.